# Relationship between wind observation accuracy and the ascending node of the sun-synchronous orbit for the Aeolus-type spaceborne Doppler wind lidar

Chuanliang Zhang[1,2], Xuejin Sun[2], Wen Lu[2], Yingni Shi[1], Naiying Dou[1], and Shaohui Li[1,2]

[1]Mailbox 5111, Beijing 100094, China

[2]College of Meteorology and Oceanography, National University of Defense Technology, Nanjing 211101, China

*Corresponding to*: Xuejin Sun (xuejin.sun@outlook.com)

**Abstract.** The launch and operation of first spaceborne Doppler wind lidar (DWL), Aeolus, is of great significance to observing global wind field. Aeolus operates on a sun-synchronous dawn-dusk orbit to minimize the negative impact of solar background radiation (SBR) on wind observation accuracy. Future spaceborne DWLs may not operate on sun-synchronous dawn-dusk orbits due to their observation purposes. The impact of the local time of ascending node (LTAN) crossing of sun-synchronous orbits on the wind observation accuracy was studied in this paper by proposing two given Aeolus-type spaceborne DWLs operated on the sun-synchronous orbits with LTANs of 15:00 and 12:00. On these two new orbits, the increments of the averaged SBR received by the new spaceborne DWLs range from 39 to 56 mW·m$^{-2}$·sr$^{-1}$·nm$^{-1}$ under cloud-free skies near the summer and winter solstices, which will lead to uncertainties of 0.19 m/s and 0.27 m/s in the increment of the averaged Rayleigh channel wind observations for 15:00 and 12:00 orbits using the instrument parameters of Aeolus with 30 measurements per observation and 20 laser pulses per measurement. This demonstrates that Aeolus operating on the sun-synchronous dawn-dusk orbit is the optimal observation scenario, and the random error caused by the SBR will be larger on other sun-synchronous orbits. Increasing the laser pulse energy of the new spaceborne DWLs is used to lower the wind observation uncertainties. And a method to quantitatively design the laser pulse energy according to the specific accuracy requirements is proposed in this study based on the relationship between the signal-to-noise ratio and the uncertainty of the response function of the Rayleigh channel. The laser pulse energies of the two new spaceborne DWLs should be set to 70 mJ based on the statistical results obtained using the method. The other instrument parameters should be the same as those of Aeolus. Based on the proposed parameters, the accuracies of about 77.19% and 74.71% of the bins of the two new spaceborne DWLs would meet the accuracy requirements of the European Space Agency (ESA) for Aeolus. These values are very close to the 76.46% accuracy of Aeolus-type spaceborne DWL when it is free of the impact of the SBR. Moreover, the averaged uncertainties of the two new spaceborne DWLs are 2.62 and 2.69 m/s, which perform better than that of Aeolus (2.77 m/s).

## 1 Introduction

The first spaceborne Doppler wind lidar (DWL) mission, the Atmospheric Dynamics Mission (ADM)-Aeolus, designed by the European Space Agency (ESA) was launched successfully on 22 August 2018. This mission has improved our knowledge

of the global wind field. Aeolus carries a spaceborne DWL, Atmospheric Laser Doppler Instrument (ALADIN), which has been used to make preliminary observations of the global wind field since its launch. Numerical Weather Prediction (NWP) experiments have shown that the assimilated wind observations have a significant positive impact on short-range wind, humidity and temperature forecasts, especially in the tropical troposphere and the South Hemisphere (Straume *et al.*, 2019).

Furthermore, scientists have also designed several possible observation scenarios for future spaceborne DWLs. For example, considering that Aeolus can only attain observations of single horizontal line-of-sight (LOS) wind components, Ma *et al.* (2015) and Masutani *et al.* (2010) proposed a spaceborne DWL concept with two pairs of telescopes (azimuth angles of one pair are 45° and 315°, and those of the other pair are 135° and 225°) using both coherent-detection and direct-detection technology; and ISHII *et al.* (2017) proposed a spaceborne coherent DWL with one pair of telescopes (azimuth angles of 45° and 315°).

Both of these observation scenarios can detect the horizontal vector wind. In addition, Marseille et al. (2008) demonstrated that a larger observation coverage is more beneficial in the improvement of NWP results on global scale compared to the measurements of the horizontal vector wind by proposing several multi-satellite joint observation scenarios with Aeolus-type instruments. Regarding multi-satellite joint observation scenarios, according to the World Meteorological Organization's (WMO) Observing Systems Capability Analysis and Review Tool (OSCAR) (Eyre, 2009), an observation cycle of 12 h with

Aeolus operating on a sun-synchronous dawn-dusk orbit would meet "the minimum" requirements that have to be met to ensure the observations are useful for global NWP. When another Aeolus-type satellite operates on a sun-synchronous noon-midnight orbit combined with Aeolus, the observation cycle may become 6 h, which would meet breakthrough requirement that, if achieved, would result in a significant improvement in global NWP compared with those based on dawn-dusk Aeolus.

Aeolus operates on a sun-synchronous dawn-dusk orbit to minimize the impact of the solar background radiation (SBR)
on the accuracy of the wind observations (Heliere *et al.*, 2002; Baars *et al.*, 2019). In this study, SBR is defined as the top-of-atmosphere (TOA) radiance that is directed toward the telescopes of the spaceborne DWL; and the solar background noise (SBN) is the photon counts excited by the SBR and imaged by the photon detectors (Zhang *et al.*, 2018), which lowers the observation accuracy due to the Poisson noise (Liu *et al.*, 2006; Hasinoff *et al.*, 2010). The dawn-dusk orbit is considered to be optimal for lowering the impact of the SBR on spaceborne DWLs operating on sun-synchronous orbits. Future spaceborne

DWLs may operate on different orbits according to their observation purposes. According to experience gained from scatterometers used in global NWP (Stoffelen *et al.*, 2013), it has been demonstrated that the forecasting errors of tropical cyclone positions are much lower when the Indian Space Research Organisation's (ISRO) scatterometer, which has an ~12:00 UTC local overpass time, is assimilated in the NWP with the original METOP-A and METOP-B (~9:30 UTC local overpass time). Therefore, it is assumed that if the global wind field at about 00:00/12:00 or 03:00/15:00 can also be observed, the global

forecast may also be significantly improved. However, if the future spaceborne DWLs operate on the sun-synchronous orbits and the local time of ascending node (LTAN) crossing differ with Aeolus, the received SBR would become larger, which would lead to higher uncertainties of the wind observations.

Aeolus is a direct-detection Doppler wind lidar that senses the winds through a Mie channel and a Rayleigh channel. According to the technology mechanism of Aeolus, the factors that affect the accuracy of the wind observations of spaceborne DWLs include atmospheric heterogeneity and SBR. The atmospheric heterogeneity mainly affects the wind observations of the Mie channel, which senses the wind using the laser signal backscattered from the aerosol/cloud particles. Sun *et al.* (2014) reported that typical values for wind uncertainties for the Mie channel in the free troposphere caused by atmospheric heterogeneity are in the range of 1−1.5 m/s, which cannot be easily corrected. For the Rayleigh channel, the uncertainties caused by atmospheric heterogeneity are 0.2–0.6 m/s in the troposphere, which can be largely reduced using a scene classification algorithm. The SBR mainly affects the observations obtained by the Rayleigh channel, which senses the wind using molecular backscatter signals. The SBR has less impact on the observations obtained by the Mie channel (Rennie, 2017). Zhang *et al.*, (2019) demonstrated that the received SBR of Aeolus ranges from 0 to 169 mW·m$^{-2}$·sr$^{-1}$·nm$^{-1}$. When the SBR is greater than 80 mW·m$^{-2}$·sr$^{-1}$·nm$^{-1}$, the entire wind observation profile is less accurate.

Observations of the global winds would improve the results of NWPs. However, the assimilation of low accuracy observations has a negative impact on the NWP results (Stoffelen *et al.*, 2005, 2006). According to the accuracy requirements of the ESA, the uncertainties of the horizontally projected line-of-sight (HLOS) wind observations in the Planetary Boundary Layer (PBL), free troposphere, and stratosphere should be less than 1, 2, and 3 m/s, respectively (Stoffelen et al., 2005). The latest research has also demonstrated that the uncertainties of 1 m/s in the PBL, 2.5 m/s in the free troposphere, and 3−5 m/s in the stratosphere would also have significant positive impacts on the NWP results (Straume *et al.*, 2019). The heights of the boundaries between the PBL, free troposphere, and stratosphere are 2 km and 16 km, respectively. In this paper, we assume that an accuracy of 5 m/s in the stratosphere is required. The free troposphere is hereinafter referred to as the troposphere.

Assuming that future Aeolus-type spaceborne DWLs would operate on sun-synchronous orbits with different LTANs, the distributions of the received SBR near the winter and summer solstices and the corresponding uncertainties of the wind observations caused by the SBR were determined in this paper. A method of lowering the uncertainty to a specific accuracy level, that is. to meet the accuracy requirements of the ESA, or to reach an accuracy level similar to that of Aeolus was also developed. In general, the only way to reduce the effect of the Poisson noise is to capture more signal (Vahlbruch *et al.*, 2008). According to the Lidar equation, the following methods can be used to increase the return signal energy of spaceborne DWLs: 1) increasing the laser pulse energy; 2) lowering the height of the orbits; 3) enlarging the telescope aperture; and 4) reducing the vertical resolution (Marseille and Stoffelen, 2003). The orbit height of Aeolus was adjusted from the originally designed 400 km to 320 km to increase the energy of the received signal. In this paper, the laser pulse energies were increased to capture more signal. The remainder of this paper is organized as follows. The details of the orbits of the three spaceborne DWLs and the Aeolus-type spaceborne DWL simulation system are presented in Section 2. Section 3 describes the method of quantitatively designing the laser pulse energy of spaceborne DWLs based on specific accuracy requirements. Before this, the relationship between the signal-to-noise ratio (SNR) and the uncertainty of the response function of the Rayleigh channel are

discussed. In Section 4, a preliminary proposal for laser pulse energies of the two new spaceborne DWLs is presented using

the method described in Section 3 based on three factors include: the global distributions of SBR and wind observation

uncertainties, and the accuracy requirements for spaceborne DWLs. Section 5 presents the summary and conclusions.

## 2 Sun-synchronous orbits and simulation system of spaceborne DWLs

In general, for sun-synchronous orbits, a spaceborne DWL operating on a dawn-dusk orbit (LTAN of 18:00) would receive

the minimum amount of SBR, and a spaceborne DWL operating on a noon-midnight orbit (LTAN of 12:00) would receive the

maximum amount of SBR. In order to study the impact of the orbit selection on the accuracy of the wind observations,

spaceborne DWLs operating on three sun-synchronous orbits with LTANs of 18:00, 15:00, and 12:00 are proposed. The

simulation system used to calculate the uncertainty of the wind observations is also described.

### 2.1 Sun-synchronous orbits

The three sun-synchronous orbits with LTANs of 18:00, 15:00, and 12:00 are illustrated in Fig. 1(a). Aeolus, which operating

on the sun-synchronous dawn-dusk orbit with height of 320 km, is marked in blue. The spaceborne DWL is equipped with a

single-perspective telescope, which scans at 90° with respect to the satellite track, has a slant angle of 35° versus the nadir,

and measures the profiles of the HLOS wind components. The other two spaceborne DWLs operating on sun-synchronous

orbits with LTANs of 15:00 and 12:00 are marked in yellow and red, respectively. The intersection points between the laser

beam and the Earth's surface are called the off-nadir points and are illustrate in Fig. 1(b).

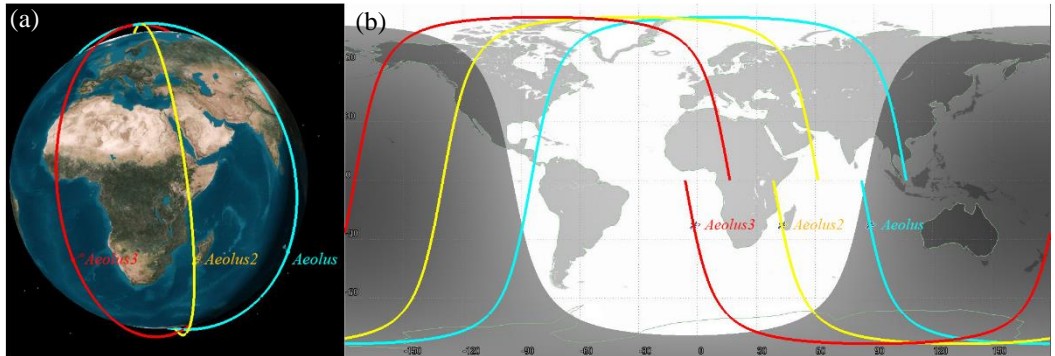

**Figure 1.** The orbits of the spaceborne DWLs operating on the sun-synchronous orbits with LTANs of 18:00, 15:00, and 12:00 are marked in blue, yellow, and red, respectively. (a) 3D graphics; (b) 2D graphics.

The two new spaceborne DWLs are assumed to be Aeolus-type instruments with the same instrument parameters as of

Aeolus, except their laser pulse energies, which are altered to improve the accuracies of their wind observations. The solar

zenith angle is the dominant factor for the SBR received by spaceborne DWLs. The variations in the solar zenith angles of the

off-nadir points on the three orbits within one year are illustrated in Fig. 2. This shows that the received SBR reaches the

maximum values near the summer and winter solstices. For the off-nadir points, in the North Hemisphere, the SBR will reach

the maximum near summer solstice, whereas it will reach the maximum near the winter solstice in the South Hemisphere. In

this paper, we focused on the observation accuracy under the worst SBR conditions. The global distributions of the maximum

SBR in a 1°×1° grid near the summer solstice (June 14 to 28) and near the winter solstice (December 15 to 30) were used for

the investigations. Furthermore, the annual variation characteristics of the solar zenith angles are less obvious for the two new

orbits compared to those of Aeolus (Fig. 2), which indicates that the observations of the two new spaceborne DWLs would

more frequently encounter the worse case SBR conditions on the Rayleigh channel compared with Aeolus.

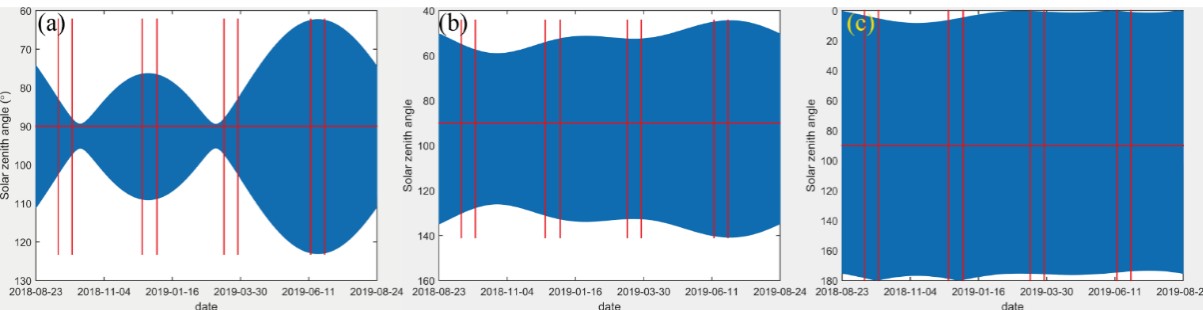


**Figure 2.** The variations in solar zenith angles of the off-nadir points on the three orbits within one year. The 4 time ranges divided by 8 red lines denote 15 days near the autumn equinox, winter solstice, spring equinox and summer solstice, respectively. Sun-synchronous orbits with LTANs of 18:00 (a), 15:00(b), and 12:00(c).

### 2.2 Spaceborne DWL simulation system

Considering the impact of the SBR on the wind observation uncertainties, an Aeolus-type spaceborne DWL simulation system

was developed to calculate the observation uncertainties. The simulation system was built according to the optical structure of

Aeolus. It consists of a laser transmitter, a telescope and front optics, a Mie spectrometer, a Rayleigh spectrometer, and front

detection units (Marseille and Stoffelen, 2003; Paffrath, 2006). Considering that the SBR mainly affects the observation

accuracy of the Rayleigh channel, we focused on the simulation of the wind retrieval method on the Rayleigh channel and

assumed that the cross-talk effect between the Mie channel and the Rayleigh channel is negligible. The details of the working

principle and the instrument parameters for Aeolus used in the simulation system, expect for the laser pulse energy, were set

according to the ADM-Aeolus Algorithm Theoretical Basis Document (ATBD) Level 1B products (Reitebuch *et al.*, 2018).

In the simulation system, one observation consists of 30 accumulations (also called 30 measurements), and one measurement

consists of 20 shots, resulting in an average horizontal length of about 90 km per observation. The detection chain noise of 4.7

e⁻/pixel on the Rayleigh channel for each measurement was also consiedered. The vertical resolutions of the retrieved wind

were 500 m in the PBL, 1 km in the troposphere, and 2 km in the stratosphere (Marseille et al., 2008).

    The input parameters of the simulation system included the u- and v- components of the wind, temperature, pressure,

aerosol optical properties, and TOA radiance. In this paper, the impacts of the SBR on the wind observation accuracy of the

spaceborne DWLs under cloudy conditions were not considered. The first five components were derived from the pseudo-

truth global atmospheric condition dataset, which consistes of the Ozone Monitoring Instrument (OMI) database (McPeters *et al.*, 2008), including the latitude-averaged profiles of the temperature, pressure, and density of ozone, and the lidar climatology

of vertical aerosol structure for spaceborne lidar simulation studies (LIVAS) database (Amiridis *et al.*, 2015), which was used

to describe the aerosol optical properties. Only the aerosols in the PBL were considered here. The details used to derive the global distributions of the SBR received by Aeolus-type spaceborne DWLs have been described by Zhang *et al.* (2019), and thus only briefly introduced here. First, the positions of the off-nadir points of the spaceborne DWLs were obtained using satellite orbit simulation software. The atmospheric conditions were retrieved from the pseudo-truth databases and were spatially interpolated to the off-nadir points. The surface albedo is also needed to generate the TOA radiance, which was derived from lambert-equivalent reflectivity (LER) database (Koelemeijer *et al.*, 2003). Then, the SBR of the off-nadir point was generated using the radiative transfer model (RTM) libRadtran with the input of atmospheric optical properties, and surface albedo (Emde *et al.*, 2016). Finally, the Earth was divided into $1° \times 1°$ grids, and the maximum SBR in each grid was selected as the worst case conditions for the Rayleigh channel wind observation uncertainties due to the SBR. Once the atmospheric conditions and SBR were input into the simulation system, the HLOS winds and their corresponding uncertainties in the grids were determined.

## 3 Methodology

In this study, a method of increasing the laser pulse energies of Aeolus-type spaceborne DWLs was developed to lower wind observation uncertainties. To assess the performance of the spaceborne DWLs under worst case conditions of the Rayleigh channel, and quantitatively design the laser pulse energies of two new spaceborne DWLs as mentioned in Section 2.1, we take the steps as follows. 1) The global distributions of the maximum SBR received by the spaceborne DWLs on the three orbits were determined. 2) The uncertainties of the wind observations of the Rayleigh channel of the three spaceborne DWLs were derived, and increments of the uncertainties of the two new spaceborne DWLs were compared to those of Aeolus. 3) The relationship between the wind observation uncertainty and the laser pulse energy was established. 4) The values of the laser pulse energies that would lower the uncertainties to the required accuracy level were derived based on the relationship established in step 3.

### 3.1 Uncertainty of the wind observations of the Rayleigh channel

The double-edge technique was used to retrieve the HLOS wind components of the Rayleigh channel for Aeolus (Flesia and Korb, 1999; Zhang et al., 2014). Tan *et al.*, (2008) showed that the uncertainty of the Rayleigh channel is determined by the response function, temperature, and pressure. A lookup table for the wind speed, response function, temperature, and pressure was established prior to the launch of Aeolus. In operation mode, the temperature and pressure profiles were obtained from the European Centre for Medium-Range Weather Forecasts' (ECMWF) data assimilation system. Once the response function of the Rayleigh channel is detected by spaceborne DWL, the wind speed can be retrieved. The uncertainty of the wind observation is estimated as

$$\sigma_{v_{HLOS}} = \frac{\partial v_{HLOS}}{\partial R_{ATM}} \sigma_{R_{ATM}},$$ (1)

where $\sigma \cdot$ is the uncertainty; $\partial \cdot$ is the partial derivative. $v_{HLOS}$ is the HLOS wind component. $R_{ATM}$ is the response function of the Rayleigh channel, which is defined as

$$R_{ATM} = \frac{N_A - N_B}{N_A + N_B},$$ (2)

where $N_A$ and $N_B$ are the useful signals detected by the Rayleigh channel.

$\partial v_{HLOS} / \partial R_{ATM}$ is a function of temperature and pressure, and it ranges from 420 to 520 m/s on most occasions (Fig. 1 of Zhang et al., 2019). The uncertainty of the response function is

$$\sigma_{R_{ATM}} = \frac{2}{(N_A + N_B)^2} \sqrt{N_B^2 \sigma_A^2 + N_A^2 \sigma_B^2},$$ (3)

where $\sigma_A$ and $\sigma_B$ are the uncertainties of $N_A$ and $N_B$, respectively. Here, $N_A$ and $N_B$ can be obtained using the simulation system of the spaceborne DWLs. Taking the SBR and the noise of the spaceborne DWL detectors into account, according to the features of the Poisson noise, the uncertainties in $N_A$ and $N_B$ can be estimated as follow:

$$\sigma_A^2 = N_A + N_{S,A} + N_{noise}^2, \text{ and } \sigma_B^2 = N_B + N_{S,B} + N_{noise}^2,$$ (4)

where the $N_{S,A}$ and $N_{S,B}$ are the photon counts excited by the SBR for the Rayleigh channel. $N_{noise}$ is the noise of the detection unit for Rayleigh channel.

$N_{S,A}$ and $N_{S,B}$ can be derived using the following method. The SBR is viewed as a spectrum with a uniform distribution, and its energy can be obtained using Eq. (5) (Nakajima *et al.*, 1999). The bandwidth is equal to that of the interference filter of the Rayleigh channel. $N_{S,A}$ and $N_{S,B}$ can be obtained from the simulation system when the spectrum is input.

$$S_{SBR} = n E_Q E_O L_S \varphi_R \frac{A_r^2 \cdot \pi}{4} \Delta\lambda \Delta t,$$ (5)

where $S_{SBR}$ is the energy of the SBR; $n$ is the number of accumulated laser shots; $E_Q$ and $E_O$ are the quantum efficiency of the detector of the Rayleigh channel (Reitebuch *et al.*, 2018); and $L_S$ is the TOA radiance of the off-nadir point. As for the instrument parameters, $\varphi_R$ is the field of view; $A_r$ is the diameter of the telescope; and $\Delta\lambda$ is the bandwidth of the interference filter. $\Delta t$ denotes the laser detection time, which is dependent on the vertical resolution.

### 3.2 Relationship between uncertainty and laser pulse energy

The laser pulse energy of the laser transmitter has an important influence on the uncertainty of the wind observation. Provided that the atmospheric conditions remain unchanged, the higher the laser energy, the stronger the backscattered signal received by the telescope of the Aeolus-type instrument, and the smaller the influence of the corresponding Poisson noise, which will finally lower the uncertainty of the wind observations. However, the quantitative relationship between the laser pulse energy and the wind observation uncertainty has not yet derived due to the fact that the wind observation uncertainties are affected by various factors such as the atmospheric conditions and instrument parameters. In this study, a method of quantitatively deriving

of the laser pulse energy according to the specific wind observation accuracy requirements is developed by establishing the relationship between the SNR of the Rayleigh channel and the uncertainty of the response function of the Rayleigh channel.

According to the characteristics of the Poisson noise, Marseille and Stoffelen, (2003) defined the SNR of the Rayleigh channel:

$$SNR_{Ray} = \frac{N_A + N_B}{\sqrt{N_A + N_B + N_{S,A} + N_{S,B} + 2N_{noise}^2}}. \tag{6}$$

For the Rayleigh channel of a spaceborne DWL, the difference between $N_A$ and $N_B$ is not large, especially when the wind speed is close to zero, that is, $N_A \approx N_B$. Based on the assumption that $N_A \approx N_B$ and $N_{S,A} \approx N_{S,B}$, we derived the relationship between the SNR and the uncertainty of the response function of the Rayleigh channel:

$$\sigma_{R_{ATM}} \approx \frac{1}{SNR_{Ray}}. \tag{7}$$

The details of the derivations and the proofs are presented in the Appendix. Then, the uncertainty of the wind observations from the Rayleigh channel can be estimated as

$$\sigma_{v_{HLOS}} \approx \frac{\partial v_{HLOS}}{\partial R_{ATM}} \cdot \frac{1}{SNR_{Ray}}. \tag{8}$$

When the laser pulse energy is increased, the value of $N_A + N_B$ will increase proportionally. Similarly, $N_{S,A} + N_{S,B}$ will increase proportionally as the SBR increases, which can be written as

$$E_{laser} \propto N_A + N_B, S_{SBR} \propto N_{S,A} + N_{S,B}. \tag{9}$$

According to Eqs. (6) and (8), $x = N_A + N_B$, which is in proportional to the energy of the laser pulse $E_{lasr}$; $y = N_{S,A} + N_{S,B}$, which is in proportional to the energy of the SBR $S_{SBR}$; and $z = \sigma_{HLOS}$. $f(T,P) = \partial v_{HLOS}/\partial R_{ATM}$ and $C = 2N_{noise}^2$, where $T$ is temperature and $P$ is pressure. Thus, the relationship between $x$, $y$, and $z$ can be expressed as

$$z \approx f(T,P) \frac{\sqrt{x+y+C}}{x}. \tag{10}$$

Equation (10) can be solved as follows:

$$x \approx \frac{f^2(T,P) + f(T,P) \cdot \sqrt{f^2(T,P) + 4z^2(y+C)}}{2z^2}. \tag{11}$$

Equation (10) illustrates that the uncertainty is determined by temperature, pressure, variable $x$, the SBR, and dark noise of the detector. The value of $x$ can be estimated using Eq. (11). Knowing the value of $x$, the value of the laser energy cannot be determined because the variable $x$ is dependent on the laser energy and the wind speed. However, when the wind speed remains unchanged, the variable $x$ is proportional to the energy of the laser pulse $E_{laser}$. That is, if the laser energy increases by several times, the corresponding value of the variable $x$ will increase by the same multiple when the HLOS wind speed remains unchanged. Then, the required value of the laser energy can be obtained based on the proportional relationship between $x$ and $E_{laser}$.

### 3.3 Derivation of laser pulse energy

In Section 3.2 and the Appendix, the relationship between the laser pulse energy $E_{laser}$ and the wind observation uncertainty was established based on several assumption and simplifications. The following method was used to solve the problem of how high to set the laser energy to increase the accuracy of the observations of the new spaceborne DWLs to the meet specific accuracy requirements.

First, the laser pulse energies of the two new spaceborne DWLs were assumed to be 60 mJ, and the parameters are the same as those of Aeolus. The profiles of the uncertainties were derived using simulation system based on the global distributions of the maximum SBR for the three orbits. Second, the profiles of variable $x$ at each bin (layer, the concept refers to Fig. 5 in Tan et al., 2008) were determined using Eq. (11), and they were set as $x1$. Assuming that the accuracy requirements of the two new spaceborne DWLs are that their accuracies reach the accuracy level of Aeolus, then the uncertainties of the new spaceborne DWLs were replaced with the uncertainties of Aeolus at the same bins, and the variables of $f(T, P)$, $y$, and $C$ were kept the same. The variables $x$ were determined using Eq. (11), and they were set as $x2$. Finally, according to the proportional relationship between $x$ and the laser energy, $E_{new}/E_{Aeolus} \approx x2/x1$, the required laser pulse energy at each bin was derived. Therefore, we could determine the laser energies of the two new spaceborne DWLs according to the statistical results.

In the same way, if the accuracies of the two new spaceborne DWLs were required to meet the accuracy requirements of the ESA, we needed to replace the wind observation uncertainties when the laser energy was 60 mJ with the accuracy requirements of the ESA when calculating the values of $x2$, and the other steps are the same as above.

## 4 Results and discussion

The preliminary results of the laser pulse energies of the two new spaceborne DWLs are presented in this section. To obtain the laser pulse energies, first the global distributions of the maximum SBR of the three orbits and the corresponding wind observation uncertainties caused by the SBR are calculated. Then, the distributions of the required laser energies are obtained according to the accuracy requirements based on the method described in Section 3.3. Finally, based on these results, the proposed laser pulse energies of the two new spaceborne DWLs are presented. The global distributions of wind observation uncertainties of the three spaceborne DWLs are determined according to the new laser pulse energies. The details are provided in the following subsections.

### 4.1 Global distributions of the maximum SBR of the three orbits

The global distributions of the maximum SBR received by the spaceborne DWLs operating on the three orbits in summer and winter are shown in Fig. 3 based on the instrument parameters of Aeolus and the three orbits described in Section 2.

The contours in Fig. 3 denote the differences between the SBR of the two new orbits and sun-synchronous dawn-dusk orbit. All the values are positive, indicating that the dawn-dusk orbit is the optimal observation scenario for minimizing received SBR for Aeolus-type spaceborne DWLs operating on sun-synchronous orbits. When operating on a sun-synchronous dawn-dusk orbit, the SBR of the off-nadir points located in the Southern Hemisphere is nearly equal to zero in summer, and the SBR of the off-nadir points located in the Northern Hemisphere is nearly equal to zero in winter. For the two new orbits, almost all of the wind observations in a few areas are not affected by the SBR, which are mainly located in the regions near the Antarctic and Arctic circles. According to the contours, the order of ascending maximum SBR, the three orbits are dawn-dusk orbit, the orbit with an LTAN of 15:00, and of the orbit with an LTAN of 12:00 respectively. The closer the LTANs of the orbits are to noon, the larger the values and the area affected by the SBR become. The statistics illustrate that the average SBR values of the dawn-dusk, 15:00, and 12:00 orbits illustrated in Fig. 3 are 20.99, 60.68, and 76.36 mW·m$^{-2}$·sr$^{-1}$·nm$^{-1}$, respectively, near the summer and winter solstice periods. The averaged increments of the SBR received by new spaceborne DWLs are 60.68-20.99=39.69 mW·m$^{-2}$·sr$^{-1}$·nm$^{-1}$ and 76.36-20.99=55.37 mW·m$^{-2}$·sr$^{-1}$·nm$^{-1}$ higher than that of Aeolus

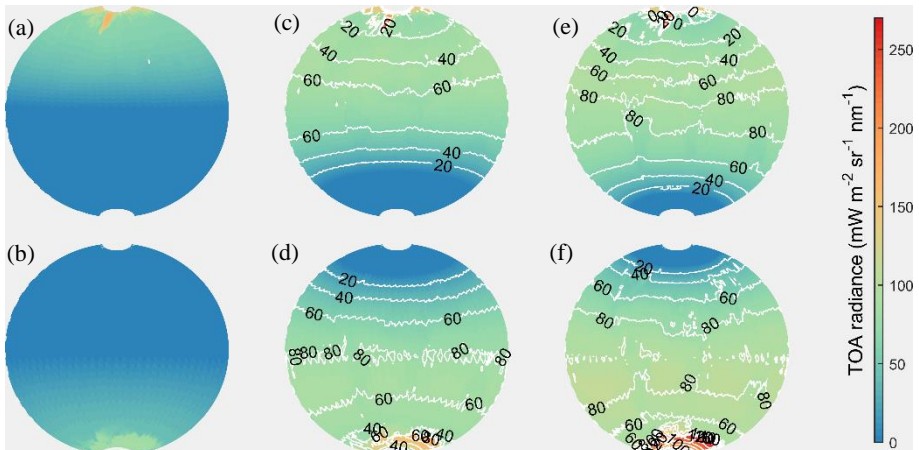

**Figure 3.** The global distributions of the maximum SBR received by spaceborne DWLs operating on the three orbits. Fig. 3(a, b), (c, d) and (e, f) present the sun-synchronous orbits with LTANs of 18:00, 15:00, and 12:00 respectively, and the upper panels denote the SBR in summer, and the lower panels denote the SBR in winter. The contours in the Fig. 3(c, e), 3(d, f) denote the differences between the SBR in Fig. 3(c, e), 3(d, f) with the SBR in Fig. 3(a, b), respectively.

**4.2 Uncertainties of wind observations based on the instrument parameters of Aeolus**

Figure 3 illustrates the global distributions of the maximum SBR near the summer and winter solstices, which focus on the worst SBR cases for the Rayleigh channel wind observation uncertainties. In fact, for sun-synchronous orbits, nearly half of the off-nadir points are in darkness, so they are free of the impact of the SBR, while the other half are in daylight and are affected by the SBR. For the off-nadir points in darkness, the latitude-averaged global distributions of the wind observation uncertainties for Aeolus-type instruments are shown in Fig. 4.

Figure 4 illustrates that 1) without the impact of the SBR, most of the wind observations in the free troposphere and stratosphere meet the accuracy requirements of the ESA. The bins for which the uncertainties exceed the requirements of the ESA are mostly located in the upper layer of troposphere and stratosphere. In addition, the accuracy of the wind observations

of Rayleigh channel in the PBL is relatively low and basically does not meet the requirements of the ESA. In fact, the Mie channel is mostly used for wind observations due to the widespread presence of aerosols in the PBL. Because the aerosols produce strong backscattered signals which can be seen as sharp peaks in the spectrum. The corresponding Doppler shifts can be determined more accurately for the spectra of sharp peaks than those of the broader molecular spectra received by Rayleigh channel. Consequently, the Mie channel wind uncertainties are smaller than those of the Rayleigh channel. Therefore, the accuracy of the Rayleigh channel in the PBL is not considered in the following section of this paper. The statistics show that the average uncertainties without the impact of the SBR are all about 2.61 m/s in troposphere and stratosphere in summer and winter overall, and about 76.46% of the bins meet the accuracy requirements of the ESA.

2) Without the impact of the SBR, the wind observation uncertainties are very similar at different latitudes.

3) The wind observation uncertainties increase with atmospheric altitude when the heights of the range gates remain unchanged. This is mainly due to the fact that the molecular number density is proportional to the pressure. Near the height of 16 km, the uncertainties decrease initially and then increase with increasing altitude, which is attributed to the change in the thickness of the bins from 1 km to 2 km.

4) Compared with other latitudinal regions, the uncertainties in the equatorial region are higher at the bottom of the troposphere and are lower in the stratosphere.

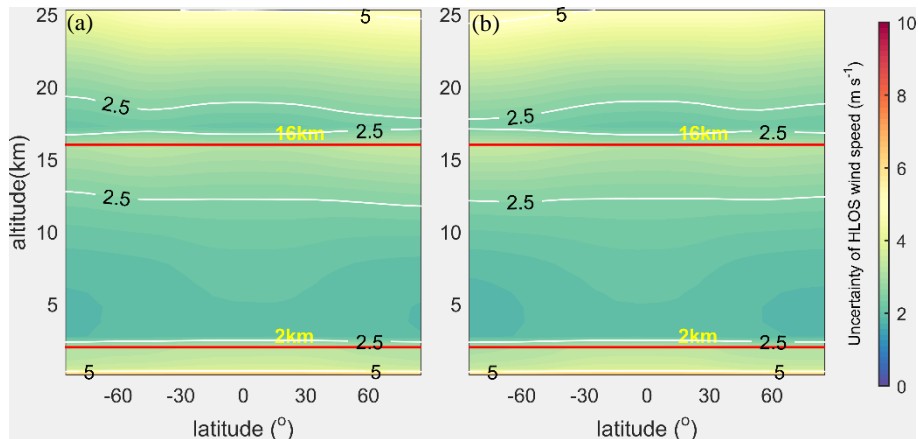

**Figure 4.** The latitude-averaged global distributions of the wind observation uncertainties without the impact of the SBR. (a) summer; (b) winter.

Based on the global distributions of the maximum SBR of the three orbits illustrated in Fig. 3, the worst SBR cases for the Rayleigh channel with maximum wind observation uncertainties were also derived (Fig. 5). Considering that the distributions of the maximum SBR are nearly horizontal to the latitudes, in to simplify the calculations, Fig. 5 was obtained using the 10 ° latitude-averaged SBR and atmospheric conditions.

As can be seen by comparing between Fig. 4 and 5, the wind observation uncertainties become larger as the impact of the SBR increases. As the LTANs of the orbits get closer to noon, the wind observation uncertainties gradually increase, and the number of bins that do not meet the accuracy requirements of the ESA also increases. For the bins in the troposphere and stratosphere, about 71.35% meet the accuracy requirements of the ESA for Aeolus; while the percentages are 63.45%

for the 15:00 orbit and 60.67% for the 12:00 orbit. The average uncertainties of the three spaceborne DWLs in the troposphere and stratosphere are 2.77, 2.96, and 3.04 m/s respectively, which illustrates that the increments of the average uncertainties of the Rayleigh channel of the new orbits are about 3.25-3.06=0.19 m/s and 3.32-3.06=0.27 m/s larger than that of Aeolus. Considering that the impact of the SBR on the wind observations is minimal on dawn-dusk orbit, and reaches the maximum on noon-midnight orbit, the selection of the LTANs of sun-synchronous orbits leads to a maximum difference

of 0.27 m/s in average global wind observation uncertainties for the Rayleigh channel of Aeolus-type DWLs near the summer and winter solstices. This small degradation of the uncertainties could also be used as an argument for operating Aeolus-type spaceborne DWLs on other sun-synchronous orbits rather than a dawn-dusk orbit, in case of flying more than a single a single Aeolus-type instrument at the same time. In addition, the average global uncertainty is 2.61 m/s without impact of the SBR as Fig. 4 indicates; and the average global uncertainty is 3.04 m/s under the worst SBR case for the

Rayleigh channel on the orbit with LTAN of 12:00. This comparison illustrates that SBR causes a maximum increase in the averaged wind observation uncertainty of about 3.04-2.61=0.43 m/s for Aeolus-type DWLs operating on sun-synchronous orbits. According to Rennie (2017), the worst case SBR (154 mW $m^{-2}\,sr^{-1}\,nm^{-1}$, polar summer condition) has noise around 0.5~1.0 m/s lager random error than the best case (night-time condition) at the height of 5~10 km. This result illustrates the degradation of the uncertainties in Rayleigh channel is not large in troposphere and also indicates the correctness of the

increase in wind observation uncertainty between different orbits calculated in this study.

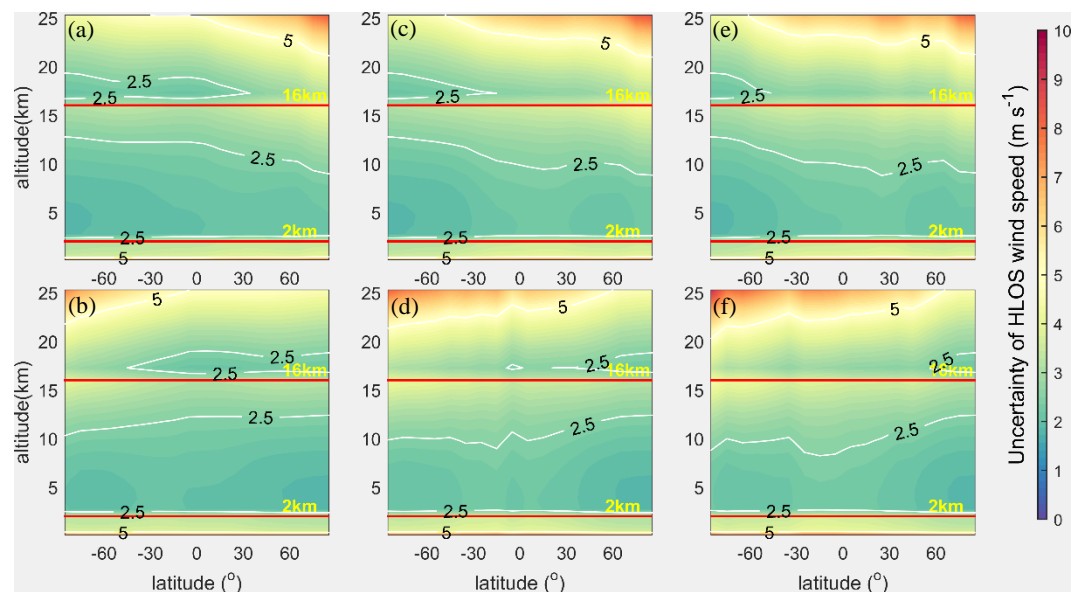

**Figure 5.** The zonal distributions of the Rayleigh channel wind uncertainties in clear air conditions observed by the three spaceborne DWLs operating on the three orbits of which the instrument parameters are the same as those of Aeolus. The contours show the accuracy requirements of ESA. The arrangement of the subgraphs corresponds to that of Fig. 2.

**4.3 Distribution of the required laser pulse energy**

In order to make the accuracies of the two new spaceborne DWLs reach the specific accuracy level for the Rayleigh channel, the required laser pulse energies were obtained using the method described in Section 3.3. According to Eq. (11), the required energy depends on the temperature, pressure, wind uncertainties, SBR, and noise of the instrument, and thus, the required laser

pulse energy is different in different bins. Therefore, the laser pulse energies of the new spaceborne DWLs should be
determined by the statistics of their required energies in different bins.

Assuming that the wind observation accuracy of the two new spaceborne DWLs needs to reach the accuracy level of the
Aeolus as is shown in Figs. 5(a, b), which can be used for joint observations of the three satellites, the global distributions of
the required laser pulse energies were derived and are illustrated in Fig. 6. Fig. 6 shows that for of the most bins of the two
new spaceborne DWLs, it is necessary to increase the laser pulse energy. Especially in the equatorial region, a higher laser
pulse energy is needed.

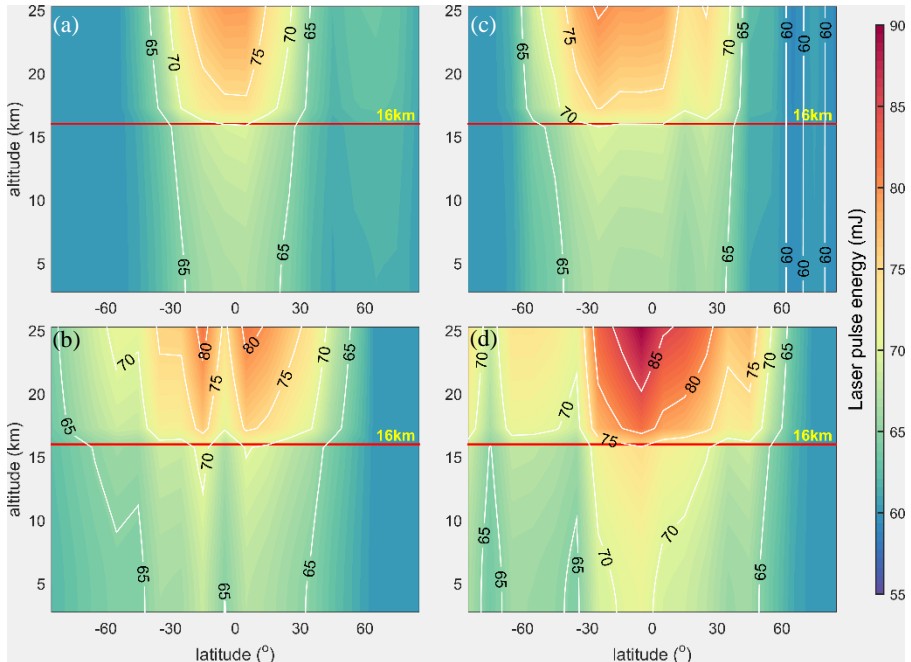

**Figure 6.** The global distributions of the required laser pulse energies in troposphere and stratosphere to make the accuracies of the two new
spaceborne DWLs reach the accuracy level of the Aeolus. Fig. 4(a, b) and (c, d) denote the sun-synchronous orbits with LTANs of 15:00
and 12:00 respectively. The upper panels denote the distributions in summer, and the lower panels denote the distributions in winter.

**Table 1.** The quantiles of the required laser pulse energies of the two new spaceborne DWLs to reach the accuracy level of Aeolus.

| Quantile (%) | | 20 | 40 | 50 | 60 | 70 | 80 | 90 | 100 |
|---|---|---|---|---|---|---|---|---|---|
| Required energy | Orbit 15:00 | 60.62 | 62.53 | 64.00 | 65.26 | 66.54 | 67.85 | 70.37 | 81.68 |
| (mJ) | Orbit 12:00 | 60.71 | 65.04 | 66.47 | 67.34 | 68.59 | 70.59 | 73.74 | 89.78 |

The statistics reveal that the average values of the required laser pulse energies in Fig. 6 are 64.80 mJ and 66.59 mJ for
the 15:00 and 12:00 orbits, respectively. The quantiles of the required energies of the two spaceborne DWLs are shown in
Table 1, which shows the corresponding percentages of the bins in which the accuracy reaches the accuracy level of Aeolus
once the laser pulse energies are equal to the specific value. For example, 90% of the bins will reach or exceed the accuracy
level of Aeolus when the laser energy is 70.37 mJ for the spaceborne DWL operating on the 15:00 orbit. As can be seen from
Table 1, when the instrument parameters of the two new spaceborne DWLs are the same as those of Aeolus, i. e., laser pulse
energies of 60 mJ, the accuracies of only about 20% of the bins reach the accuracy level of Aeolus near the summer and winter

solstices. However, when the laser energy is slightly increased, the percentage of the bins greatly increases. When the laser

pulse energy reaches 70 mJ, the accuracies of about 90% and about 80% of the bins reach or exceed the accuracy level of

Aeolus on the orbit 15:00 and 12:00 orbits, respectively.

Another potential application of the new spaceborne DWLs is to enlarge the global wind observation coverage to improve

the forecast results of NWPs. This should have a positive impact on the NWP results when the wind observation accuracy

meets the requirements of the ESA. The distributions of the laser pulse energies of the three orbits required to meet the accuracy

requirements of the ESA are illustrated in Fig. 7.

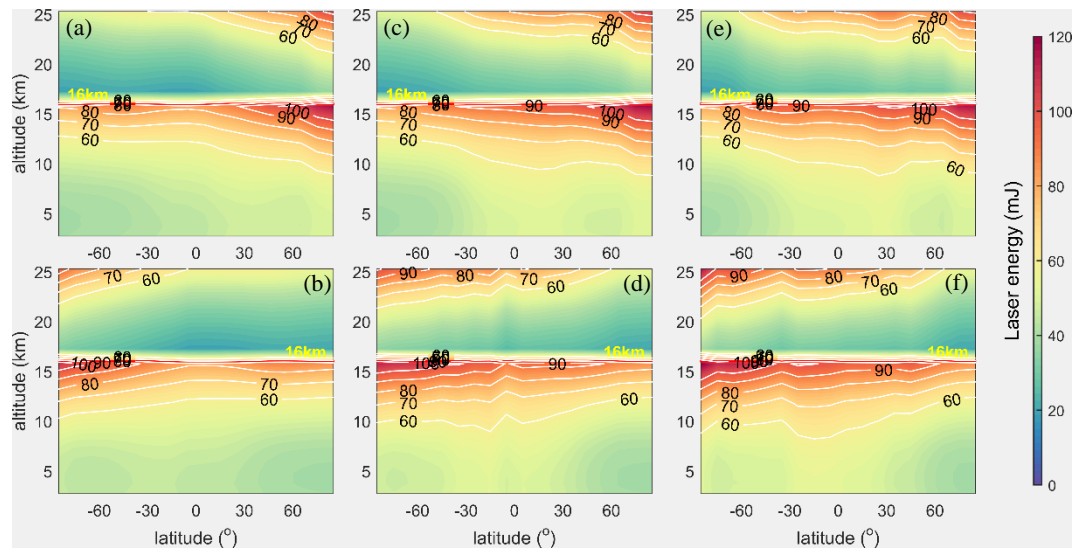


**Figure 7.** The global distributions of the required laser pulse energies in the troposphere and the stratosphere to reach the accuracy requirements of the ESA. The arrangement of the subgraphs corresponds to that of Fig. 2.

**Table 2.** The quantiles of the required laser pulse energy of the three spaceborne DWLs to meet the accuracy requirements of the ESA.

| Quantile (%) | | 20 | 40 | 50 | 60 | 70 | 80 | 90 | 100 |
|---|---|---|---|---|---|---|---|---|---|
| Required energy (mJ) | Orbit 18:00 | 40.93 | 46.33 | 49.35 | 53.87 | 59.22 | 67.89 | 78.63 | 116.96 |
| | Orbit 15:00 | 42.83 | 50.88 | 53.17 | 58.21 | 63.96 | 73.71 | 84.88 | 118.20 |
| | Orbit 12:00 | 45.06 | 51.81 | 54.76 | 59.86 | 66.46 | 75.98 | 86.82 | 121.19 |

Figure 7 illustrates that the wind observation uncertainties of most of the bins in the lower level of the troposphere and

the stratosphere meet the accuracy requirements of the ESA for the three spaceborne DWLs with a laser pulse energy of 60

mJ. Higher energies are needed in the upper level of the troposphere and the stratosphere, especially in for the regions close to

the Antarctic and Arctic circles. On the boundary line with height of 16 km, there is an obvious sudden decrease in the required

laser energies. This is mainly because the vertical thickness of the observation bins changes from 1 km in the troposphere to 2

km in the stratosphere, which doubles the integration time of the detection units of Rayleigh channel. Larger atmospheric

backscattered signal will be integrated. Moreover, the required wind observation uncertainties increase from 2 m/s to 3 m/s.

Therefore, the required laser energies suddenly decrease when transitioning from the troposphere to the stratosphere near a

height of 16 km. The comparison of the required laser energies of the three orbits illustrates that the closer the orbital LTANs

are to noon, larger the average values of the required laser energies will become. The statistics show that the average values of the required energies are 53.27 mJ for Aeolus, 57.60 mJ for the 15:00 orbit, and 59.19 mJ for the 12:00 orbit. The quantiles

of the required energies of the three spaceborne DWLs are shown in Table 2. The statistics presented in Table 2 illustrate that the percentages of the bins that meet the accuracy requirements of the ESA increase by 10% even if the laser pulse energy is not increased significantly when the quantile is between 40% to 90%. The average increment of the laser pulse energy is 6.75 mJ which can increase the quantiles by 10% for the three orbits as a whole. When the laser pulse energies are set to 67.89, 73.71, and 75.98 mJ, the quantiles are up to 80%, which exceeds the percentage of bins (76.46%) for Aeolus without the impact

of the SBR.

**4.4 Uncertainties of wind observations resulting from an increased laser pulse energy**

In Section 4.3, the zonal distributions of the required laser pulse energies were derived for different purposes. In order to offer a feasible proposal for the laser pulse energies of the new spaceborne DWLs, and the percentages of the bins that meet the specific accuracy requirements when the laser energies reach certain values were determined as is shown in Table 3.

Considering the accuracy requirements of the ESA, the accuracy level of Aeolus, and taking the existing technical level into account, the laser energies of the two new spaceborne DWLs were set to 70 mJ in this study. In fact, the laser energy of 80 mJ has already been required by the ESA in the ATBD (Reitebuch et al., 2018), and it has been achieved in the initial orbiting phase of the satellite. As is shown in Table 3, the percentages of the bins that meet the accuracy requirements of the ESA are 77.19% and 74.71% for the 15:00 and 12:00 orbits, respectively, which are close to the percentage of Aeolus without

the impact of the SBR (76.46%). The percentages of the bins are up to 89.04% and 77.34% for the 15:00 and 12:00 orbits, of which bins the accuracy are equals to or exceed the accuracy level of Aeolus.

**Table 3.** Percentages of bins which will meet the specific accuracy requirements with certain laser pulse energies for spaceborne DWLs.

| Accuracy requirements | | Laser pulse energy (mJ) | | | | | |
|---|---|---|---|---|---|---|---|
| | | 50 | 60 | 70 | 80 | 90 | 100 |
| ESA (%)[a] | Orbit 18:00 | 51.61 | 71.35 | 82.89 | 90.50 | 96.64 | 98.54 |
| | Orbit 15:00 | 37.13 | 63.45 | 77.19 | 85.53 | 93.42 | 97.66 |
| | Orbit 12:00 | 33.33 | 60.67 | 74.71 | 84.21 | 91.96 | 97.22 |
| Aeolus (%)[b] | Orbit 15:00 | 0 | 19.44 | 89.04 | 99.42 | 100 | 100 |
| | Orbit 12:00 | 0 | 16.67 | 77.34 | 96.78 | 100 | 100 |

[a] The percentage of bins which will meet the accuracy requirements of ESA when the laser energies reach the specific value.

[b] The percentage of bins which will reach the accuracy level of Aeolus in the corresponding bins when the laser energies reach the specific

value.

For the three spaceborne DWLs operate on the sun-synchronous orbits shown in Fig. 1, the instrument parameters of Aeolus remain unchanged. As to the two new Aeolus-type spaceborne DWLs, the other instrument parameters are set the same as those of Aeolus, except for the laser pulse energies of 70 mJ. The wind observation uncertainty distributions of the three spaceborne DWLs were derived and are shown in Fig. 8. Note that Fig. 8(a, b) is identical to Fig. 5(a, b), because both were

obtained with laser energies of 60 mJ.

As is illustrated by Table 3, when the laser pulse energies of the dawn-dusk, 15:00, and 12:00 spaceborne DWLs are 60, 70, and 70 mJ, respectively, the percentages of the bins that meet the accuracy requirements of the ESA are close (71.35%, 77.19%, and 74.71%, respectively). And Fig. 8 illustrates that the bins that reach the ESA's accuracy requirements have very consistent latitude and height distributions for the three orbits. By Comparing among Fig. 8(c−f) and Fig. 4, it can be seen that

the wind observation accuracy is significantly improved in the hemisphere that is less affect by the SBR. However, limited improvement occurs in the other hemisphere. This indicates that increasing the laser energy to 70 mJ cannot completely compensate for the negative influence of the large amount of the SBR. However, by Comparing Fig. 8(c−f) and Fig. 5(c−f), it can be seen that the wind observation accuracy is still greatly improved when the laser pulse energy is increased from 60 mJ to 70 mJ. The fact that such improvements are obtained for only a 10 mJ increase in the laser pulse energy illustrates that the

wind observation uncertainties are sensitive to the laser pulse energy of the spaceborne DWLs. The average uncertainties of the two new spaceborne DWLs with a laser pulse energy of 70 mJ in troposphere and stratosphere are 2.62 and 2.69 m/s respectively. Compared to the average uncertainties for a laser pulse energy of 60 mJ, the difference in the uncertainties are 2.96−2.62=0.34 m/s and 3.04−2.69=0.35 m/s smaller, which indicates that when the laser pulse energies of the two new spaceborne DWLs are increased from 60 mJ to 70 mJ, the global average wind observation uncertainties decrease by about

0.34 m/s under the impact of the maximum SBR conditions.

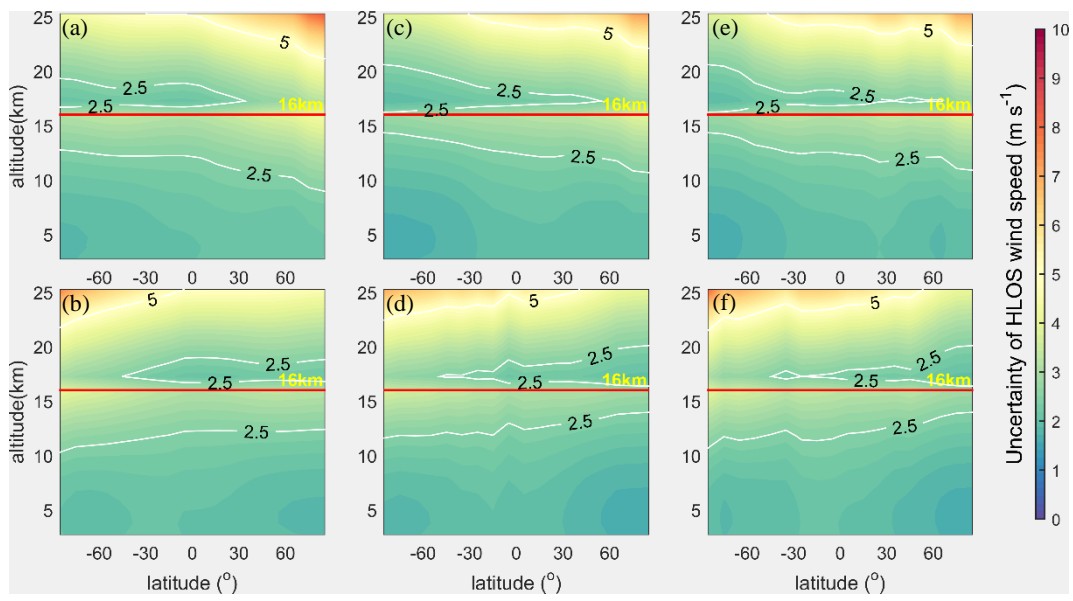

**Figure 8.** The zonal distributions of the wind observation uncertainties of the three spaceborne DWLs with the laser energy of 60 mJ for Aeolus , and with a laser energy of 70 mJ for the two new Aeolus-type spaceborne DWLs. The arrangement of the subgraphs corresponds to that of Fig. 2.

**5 Summary and conclusions**

The successful launch of Aeolus is significant for observing the global wind field. Aeolus operates on the sun-synchronous dawn-dusk orbit to minimize the impact of the SBR on the accuracy of the wind observations. If future spaceborne DWLs operate on other sun-synchronous orbits to fulfil their specific observation purposes, the received SBR may become larger, which would lead to higher observation uncertainties. In general, for sun-synchronous orbits, a spaceborne DWL operating on

a dawn-dusk orbit (LTAN of 18:00) will receive the minimum SBR; and a spaceborne DWL operating on a noon-midnight orbit (LTAN of 12:00) will receive the maximum SBR. In this paper, the influence of the LTAN of the sun-synchronous orbit on the wind observation accuracy of Aeolus-type spaceborne DWLs was investigated based on two spaceborne DWLs operating on sun-synchronous orbits with LTANs of 15:00 and 12:00 combined with Aeolus. The method of increasing the laser pulse energies of spaceborne DWLs was used to lower the observation uncertainties. Furthermore, a method of

quantitatively designing laser pulse energy to meet specific accuracy requirements was developed.

For two new Aeolus-type spaceborne DWLs operating on sun-synchronous orbits with LTANs of 15:00 and 12:00, the global distributions of the SBR illustrate that the increments of the average SBR range from 39 to 56 $mW \cdot m^{-2} \cdot sr^{-1} \cdot nm^{-1}$ near the summer and winter solstices compared to that of the Aeolus under cloud-free skies. This leads to the average uncertainty increments of 0.19 m/s and 0.27 m/s for the 15:00 12:00 orbits, respectively. Considering that the impact of the SBR on the

wind observations is minimal on a dawn-dusk orbit, and reaches the maximum on a noon-midnight orbit, the selection of the LTAN of a sun-synchronous orbit will result in a maximum difference of 0.27 m/s in the global average wind observation uncertainties for the Rayleigh channel of Aeolus-type DWLs near the summer and winter solstices. Furthermore, the average global uncertainty is 2.61 m/s without the impact of the SBR, and the average global uncertainty is 3.04 m/s under the worst SBR case for Rayleigh channel on the orbit with an LTAN of 12:00. This fact illustrates that the maximum increase in the

average global wind observation uncertainty due to SBR is 3.04−2.61=0.43 m/s for Aeolus-type DWLs operating on sun-synchronous orbits. In addition, the statistics show that 71.35% of the bins of Aeolus meet the accuracy requirements of the ESA in the free troposphere and in the stratosphere near the summer and winter solstices. For the two new spaceborne DWLs, the percentages are 63.45% and 60.67% for the 15:00 and 12:00 orbits. Therefore, it is necessary to increase the laser pulse energies of the two new spaceborne DWLs to promote wind observation accuracy and to increase the percentage of bins that

meet accuracy requirements of the ESA. Moreover, the wind observation uncertainties are sensitive to the laser pulse energies, and results of this study show that the percentage of bins that meet the accuracy requirements of the ESA would increase by 10% when the laser pulse energy is increased by an average of only 6.75 mJ for the three orbits.

To quantitatively design the required laser pulse energies of the new spaceborne DWLs so that they meet the specific accuracy requirements, i.e., the accuracy requirements of the ESA, or reach the accuracy level of Aeolus, the relationship

between the SNR and the uncertainty of the response function of the Rayleigh channel was established based on several

assumption and simplifications. This is demonstrated to have a wide feasibility by simulation experiments, which is shown in the Appendix. Finally, a method of deriving the required laser energies according to the accuracy requirements is proposed.

According to this method, the required energy is based on the temperature, pressure, wind uncertainty, SBR, and noise of the instrument, and thus, the required laser pulse energies are different in different bins. Therefore, the laser pulse energies of
the spaceborne DWLs should be determined based on the statistics. In order to reach the accuracy level of Aeolus, and improve the forecast results of the NWPs, and taking the existing technical level of spaceborne DWLs into account, the laser pulse energies of two new spaceborne DWLs were set to 70 mJ, while other parameters were the same as those of Aeolus. Based on the proposed parameters, 89.04% and 77.34% of the bins reach the accuracy level of Aeolus for the 15:00 and 12:00 orbits. Moreover, the percentages of the bins that meet the ESA's accuracy requirements are 77.19% and 74.71% for the two new
spaceborne DWLs, which are higher than that of Aeolus (71.35%), and are close to the percentage for 76.46% Aeolus when it is free of the impact of the SBR. The average uncertainties of the two new spaceborne DWLs with laser pulse energies of 70 mJ in the free troposphere and stratosphere are 2.62 and 2.69 m/s, respectively, which perform better than that of Aeolus (2.77 m/s). Furthermore, when the laser pulse energies of the two new spaceborne DWLs increase from 60 mJ to 70 mJ, the average global wind observation uncertainties decrease by about 0.34 m/s under the impact of the maximum SBR. In summary, it is
necessary to increase the laser pulse energies of the two new Aeolus-type spaceborne DWLs operating on sun-synchronous orbits with LTANs of 15:00 and 12:00. The wind measurement accuracy is greatly improved when the laser pulse energies are increased from 60 mJ to 70 mJ.

The essence of lowering the wind observation uncertainties of spaceborne DWLs by increasing the laser pulse energies is to increase the SNR of the received signal. Other methods can be used to improve the SNR of the received signal, such as
enlarging the telescope aperture or reducing the vertical resolution. Once the quantitative relationship between these instrument parameters and the SNR is established, we can also quantitatively adjust these parameters according to the accuracy requirements using the method described in this paper.

**Appendix**

To build the relationship between the laser pulse energies and uncertainties of wind observations for Aeolus-type spaceborne
DWLs, we derived the relationship between the response function and the SNR of the Rayleigh channel. According to Eq. (3) and (4), the uncertainty of response function of Rayleigh channel can be written as follows based on the assumption that $N_A \approx N_B$ and $N_{S,A} \approx N_{S,B}$,

$$\begin{aligned} \sigma_{R_{ATM}} &= \frac{2}{(N_A+N_B)^2} \sqrt{N_B^2(N_A + N_{S,A} + N_{noise}^2) + N_A^2(N_B + N_{S,B} + N_{noise}^2)} \\ &\approx \frac{2}{4N_A^2} \sqrt{2N_A^2(N_A + N_{S,A} + N_{noise}^2)} \\ &= \frac{\sigma_A}{\sqrt{2}N_A} \end{aligned} \qquad (A1)$$

According to Eq. (6), the SNR of the Rayleigh channel for spaceborne DWLs can be expressed as

$$
\begin{aligned}
SNR_{Ray} &= \frac{N_A+N_B}{\sqrt{N_A+N_B+N_{S,A}+N_{S,B}+2N_{noise}^2}} \\
&\approx \frac{2N_A}{\sqrt{2(N_A+N_{S,A}+N_{noise}^2)}} \\
&= \frac{\sqrt{2}N_A}{\sigma_A}
\end{aligned}
\qquad (A2)
$$

Therefore,

$$
SNR_{Ray} \approx \frac{1}{\sigma_{R_{ATM}}}. \qquad (A3)
$$

As is the equations derivation process shown in Section 3.2, the relationship between the SNR and the uncertainty of response function shown in Eq. (A3) is the basis to derive the relationship between the laser pulse energy and the wind observation uncertainty shown in Eq. (10) and (11). However, Eq. (A3) is derived through assumption and simplifications, especially the assumption $N_A \approx N_B$, of which the values may be of large differences when the absolute values of HLOS wind speed are large. To test the correctness of Eq. (A3) in the actual atmosphere with variable wind speed, we verified the equation using reanalysis data, aerosol optical parameters database LIVAS and surface albedo database. The verification process is shown in Fig. A1.

The reanalysis data were obtained from the 20th Century Reanalysis Project (Compo *et al.*, 2011). In the validation experiments, the monthly averaged 24 level profiles of the temperature, pressure, u- and v-components of the wind with 1°×1° spatial resolutions were obtained from the reanalysis data. In this study, the reanalysis data for June 2015 and December 2015 were used as the atmospheric conditions in summer and winter, respectively. As is shown in Fig. A1, the verification process described by Eq. (A3) can be described as follows.

(1) The off-nadir points of the spaceborne DWLs are obtained using orbit simulation software based on the orbit information of the spaceborne DWLs.

(2) The profiles of the temperature, pressure, wind speed, aerosol optical parameters, and surface albedo are interpolated into the off-nadir points.

(3) The SBR values of the off-nadir points are derived using the RTM libRadtran with the inputs provided in step (2).

(4) The profile values of $N_A$, $N_B$ and $N_{S,A}$, $N_{S,B}$ are determined using spaceborne DWL simulation system described in Section 2.2 with the inputs of SBR and atmospheric conditions of the off-nadir points.

(5) The values of $\sigma_{R_{ATM}}$ and $SNR_{Ray}$ are obtained using Eqs. (3), (4), and (6). In addition, according to the ADM-Aeolus ATBD Level 1B products (Reitebuch *et al.*, 2018), the noise of the detection chain for each measurement is 4.7 e⁻/pixel. There are 30 measurements in one observation, therefore, $C = 2N_{noise}^2 = 2 \times (4.7 \times 30)^2 = 39762$ in Eq. (10), which is not negligible.

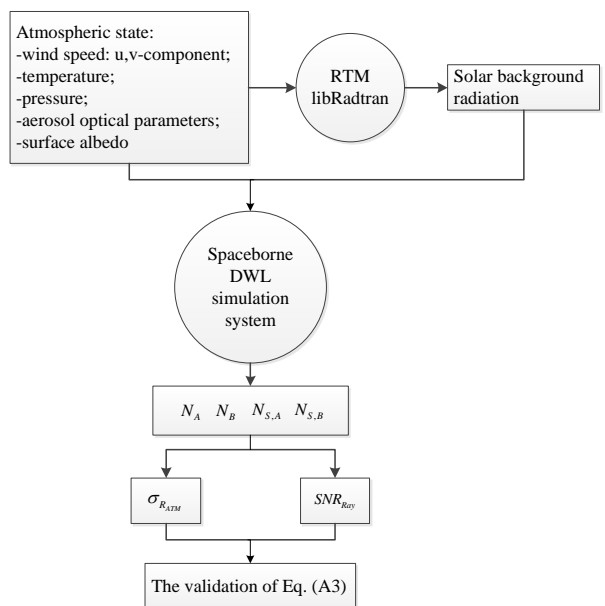

**Figure A1.** The verification process of Eq. (A3).

The scatters of $\sigma_{R_{ATM}}$ and $1/SNR_{Ray}$ are plotted to verify the accuracy of Eq. (A3), as is shown in Fig. A2. The spatial resolution of the reanalysis data is 1°×1°, so the Earth is divided into 1°×1° grids during the verification process, and one off-nadir point in each grid is selected as the verification point. Considering the SBR in summer and winter, and excluding some grid points with invalid data, a total of 28460 profiles are used in the verification. Each profile contains 24 bins, and the verification uses 683040 scattered points.

In the verification, the HLOS wind components derived from u- and v-wind components range from -73.02 to 33.14 m/s. Fig. A2 illustrates that the points plot of the reciprocal SNR versus the uncertainty of the response function of the Rayleigh channel plot very close to the $y = x$ line, which demonstrates that the assumption and simplifications used in deriving the relationship between the laser pulse energy and the uncertainty of the wind observation are reasonable, and Eq. (A3) has a wide applicability and feasibility in the real atmosphere.

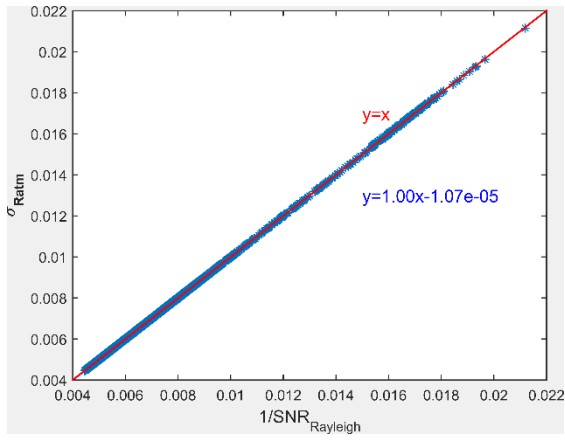

**Figure A2.** The scatter plot of the reciprocal SNR versus the uncertainty of the response function of the Rayleigh channel and their first order fitting relationship.

The variables used in the verification of Eq. (A3) can also be used in the verification of Eq. (11). The variable $\partial v_{HLOS}/\partial R_{ATM}$ is also needed. It is a function of temperature and pressure, and can be obtained through a pre-calculated lookup table. The verification results for Eq. (11) are shown in Fig. A3.

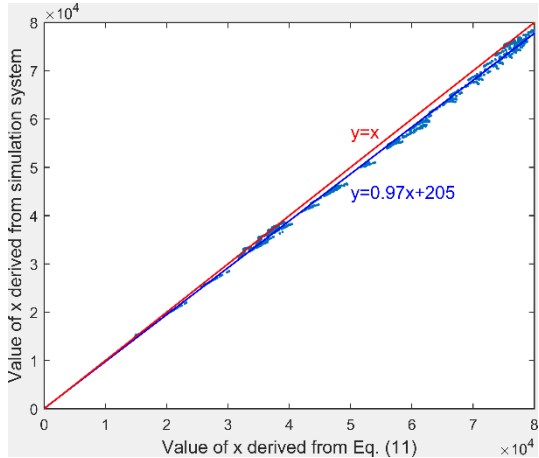

**Figure A3.** The scatter plot of the $x$ values which are derived from Eq. (11) and simulation system. And $x$ is the sum of $N_A$ and $N_B$.

As is shown in Fig. A3, the fitting line of the points on the plot of the $x$ values derived from Eq. (11) versus the values derived from the simulation system plot very close to the $y = x$ line. Furthermore, the residuals between the points and the fitting line are very small, which indicates the wide feasibility and applicability of Eq. (11). In addition, it should be noted that the points in Fig. A3 mostly plot below the $y = x$ line, which indicates that the $x$ values calculated using Eq. (11) are smaller

than the actual values. According to Section 3.3, the laser pulse energy is derived based on the equation $E_{new}/E_{Aeolus} \approx x2/x1$; and $x1$ is obtained from the simulation system, which is regarded to be close to the real value. A smaller $x2$ may lead to a smaller $E_{new}$, which is about 0.97 times the real value.

*Code and Data availability.* The codes in this article are mainly compiled using matlab and are available upon request from

the first author by email, zhang01020@hotmail.com. The databases used in this paper include: OMI database, which provided the latitude-averaged temperature, pressure, and ozone, can be accessed via anonymous ftp from toms.gsfc.nasa.gov/ pub/LLM_climatology; LIVAS database, providing the golabl aerosol optical properties with 1°×1° grid, offered by Dr. V. Amiridis from Instiude for space applications and remote sensing, National observatory of Athens, and can be assessed from http://lidar.space.no a.gr:8080/livas/; the global LER database is available upon request from the authors, Dr. R. B. A.

Koelemeijer from Air Research Laboratory, National Institute of Public Health and the Environment, robert.koelemeijer@rivm.nl; and the reanalysis data of 20th Century Reanalysis provided by the NOAA/OAR/ESRL PSD, Boulder, Colorado, USA, from their Web site at https://www.esrl.noaa.gov/psd/.

*Author contributions.* CZ, XS, and WL designed the studies; CZ built the simulation systems, performed the computation and analysis, and wrote the paper text; YS, ND, and SL provided important information on data delivery and processing. All authors engaged in discussions on studies, interpretation of results, as well as contribution to the finalization of the paper text.

*Competing interests.* The authors declare that they have no conflict of interest.

*Acknowledgements.* Thanks for the helpful discussions provided by Dr. Karsten Schmidt from DLR, Dr. Gert-Jan Marseille and Dr. Ad Stoffelen from Royal Netherlands Meteorological Institute in the building simulation system of Aeolus-type spaceborne DWLs. Thanks for the suggestions provided by Dr. Claudia Emde in running the libRadtran.

*Financial support.* This research was supported by National Natural Science Foundation of China (NSFC) (41575020).

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
