# Peer review of "Relationship between wind observation accuracy and the ascending node of the sun-synchronous orbit for the Aeolus-type spaceborne Doppler wind lidar"

_Atmospheric Measurement Techniques, 2020_

## Referee Comment (RC1) · Gert-Jan Marseille (Referee) · 13 Jul 2020

The authors provide suggestions for future Aeolus-type follow-on missions, with different local overpass times compared to Aeolus' 6/18 UTC (dawn-dusk) and taking into account increased solar background radiation in measured Aeolus signals, hence reduced data quality, as a consequence of selecting different sun-synchronous orbits.

Main comments ===============

The presented results are interesting and potentially useful input for discussions on an

[Figure]

Aeolus follow-on mission. The main question, which has not been answered in the paper is: why selecting orbits other than dawn-dusk, given that 3 satellites in a dawn-dusk orbit gives already quite good global coverage, without reducing wind quality due to increased solar background? See Fig.4 of Marseille et al (2008). Is there some indication that different local overpass times would be favorable for NWP? Please elaborate on this in the paper.

In connection to this. Line 43: "The future spaceborne DWLs may operate on different orbits which should be related to their observation purposes". Which observation purposes are related to 12:00 or 15:00 local overpass times? See also line 72: "Assuming the future Aeolus-type spaceborne DWLs will operate on the sun-synchronous orbits with different LTAN". Based on what assumption?

The realism of the simulations can be largely improved by comparing Aeolus measured SBR with simulated Aeolus SBR. Information on Aeolus measured SBR and the impact on Aeolus wind quality is found in the attached supplementary material. Based on this information, the authors can test the realism of their simulations.

Minor comments ============

line 60: "The received SBR of Aeolus ranges from 0 to 169 mWâŃĚm−2âŃĚsr−1âŃĚnm−1". That is worse than "On the two new orbits, the increments of averaged SBR received by the new spaceborne DWLs range from 39 to 56 mWâŃĚm−2âŃĚsr−1âŃĚnm−1" (line 14). Can the author please comment.

Figure 1b is misleading since it suggests that all three Aeolus-type instruments operate during daytime at equal time intervals.

line 108: "Figure 1(b) shows that the solar zenith angle of the observation points of the two new Aeolus-type instruments is low compared to that of Aeolus". How can that be seen from the figure?

The simulations would be more useful if the operational Aeolus instrument would be

used for reference 1. one measurement is composed of 20 accumulated shots on-board; 1 observation is obtained from averaging 30 measurements. 2. Assuming around 60mJ laser energy, which is consistent with current operational Aeolus laser-B 3. optical throughput is a factor 2-3 lower than expected.

The authors can do simulations based on both this unexpected signal loss (worst case scenario) and without this loss, assuming that the problem can be identified and solved before the launch of the Aeolus follow-on mission (best case scenario). With the above settings 0% of Aeolus data would meet the mission requirement, rather than 88.01% as mentioned in Table 5. So, it would be interesting to extend Table 5, by presenting both best and worst case scenarios.

line 118: "we focus on the simulation of the wind retrieved method on Rayleigh channel, and assume that the cross-talk effect between Mie channel and Rayleigh channel is negligible". Based on this assumption, you can remove mentioning over scattering ratio in section 3.1.

Derivation of Eq. (7) in Appendix could have been done more simple, by substituting A=B in Eq.(3) => sigma_R_ATM = sigma_a/(N_a*sqrt(2)) Substituting Eq. (4) in Eq.(6) and setting A=B => SNR_Ray = (N_a*sqrt(2))/sigma_a

Figure 2. For the 15:00 and 12:00 UTC orbits, half the orbit is in full darkness (so no SBR contribution), the other half in full daylight (a large SBR contribution). How is this reflected in figure 2?

Also in Figure 3, I would expect bi-modal accuracy statistics with very good quality at the dark part of the orbit (no SBR) and low quality in the day-light part (high SBR). So, what is exactly displayed in Figure 3? Please present both statistics separately. In the caption of figure 3, mention that this is winds from the Rayleigh channel in clean air conditions.

Please also note the supplement to this comment:

[Figure]

https://www.atmos-meas-tech-discuss.net/amt-2020-202/amt-2020-202-RC1-supplement.pdf

[Figure]

**Supplement:**

**What drives the random errors?**

1. **Laser emit energy**

   ⇨ Lower than expected (factor 1-2)

   ⇨ Negative trend

2. **Optical signal throughput** in receive path for atmospheric signal

   ⇨ Lower than expected (factor 2-3)

   ⇨ Negative trend

3. **Solar background noise**

   ⇨ Impact higher than expected due to lower atmospheric signal

   ⇨ Seasonal variation of solar background by factor 18: Rayleigh random errors of 7-8 m/s were obtained in summer months for polar regions

Figures by **K. Schmidt (DLR)**.

**Orbital variation of Rayleigh solar background noise**

[Figure]

[Figure]

**Rayleigh winds are very sensitive to solar background noise with current *low useful signal levels**

[Figure]

This is a simulation, but tuned to actual L2B Rayleigh-clear random errors found

Given our current useful signal **we have a lot to gain in wind random error** from more signal, particularly in polar summer conditions

Curve with **worst case** solar background noise e.g. polar summer

Curve with **typical** solar background noise

Unfortunately, losing signal means noise increases a lot

*Recent* signal level for 14 km altitude, 1 km range-bin

Doubling signal would massively reduce Rayleigh wind noise

Rayleigh wind error, HLOS (m/s)

Useful signal level on channel A (at BRC level)

7 m/s
6 m/s
4 m/s
3.9 m/s
2.7 m/s

---

## Referee Comment (RC2) · Karsten Schmidt (Referee) · 18 Aug 2020

general comments

ESA's Earth Explorer Aeolus is the first satellite in space which measures globally wind profiles by use of a Doppler Wind Lidar (DWL). Due to its success, different scenarios of a follow-on mission are currently discussed. The paper contributes to this discussion. Sun-synchronous orbits with local time of ascending node (LTAN) of 15:00 and 12:00 of two additional Aeolus-type spaceborne DWLs are considered. The solar back-

ground radiation (SBR), seen by these satellites, is computed. It is found that SBR is increased due to the choice of the orbits. As a result, also increased Rayleigh channel wind errors, compared to Aeolus, are obtained for the two new satellites at cloud-free atmospheric conditions. The influence of an increased laser pulse energy of 80mJ is investigated to compensate the larger Rayleigh wind errors. In particular, a scheme is derived and applied to quantitatively design the required laser pulse energy of the new DWLs to meet specific accuracy requirements. Thus the paper addresses relevant scientific questions within the scope of AMT. It is of interest for the scientific community and should be published in AMT. However, there are some points which should be considered by the authors.

major specific comments

(1) The authors should compare SBR computed by means of their model with measured in-orbit SBR data for certain time ranges. The reviewer 1 provided some data in his review supplement. This would increase the confidence in the authors model. Moreover, plots over a year show that SBR measured by Aeolus is maximum in June and December (see again supplement). Thus the authors can argue that their investigations for June and December are for the worst cases with maximum Rayleigh channel wind errors due to SBR.

(2) In lines 60-61, the authors write that the "received SBR of Aeolus ranges from 0 to 169 mW*mˆ-2*srˆ-1*nmˆ-1". The authors should give the corresponding reference. Aeolus measures primarily ACCD counts of SBR.

(3) It is of course possible and interesting to consider sun-synchronous orbits other than dawn-dusk orbits. The authors should explain their choice of orbits with LTANs of 15:00 and 12:00 in Section 2.1.

(4) It becomes not clear which kinds of aerosols are considered by the authors in their simulations (only aerosols in the planetary boundary layer (PBL) or also above it). The authors should specify this. Furthermore, the authors should replace "clear sky" by

"cloud-free" in lines 15 and 379 due to the presence of aerosols.

(5) For the simulations, the Aeolus instrument parameters have been taken by the authors from the Algorithm Theoretical Basis Document (ATBD; Reitebuch et al., 2006). There is however a newer version of this document (issue 4.4, 20.04.2018), e.g. available by ESA for Aeolus CalVal users. Furthermore, the authors considered observations consisting of 50 accumulations (measurements) of 14 shots, resulting in a horizontal resolution of about 100.8 km per observation. However, Aeolus has 30 measurements per observation with 20 laser pulses per measurement (in the level 1B processing), resulting in a horizontal averaging length of about 90km per observation. So the averaged wind observation uncertainties, derived by the authors in the present study, are only some estimates. It is proposed to use the newer/current parameters in future simulations in order to increase their usefulness.

(6) Eq. (7) has been numerically verified in the Appendix by neglecting noise (see item (5) in line 436). Consequently, Eq. (8) holds only for this restriction. Then, Eq. (8) is reformulated to Eq. (10) by using Eq. (6). However, Eq. (6) does contain noise, and consequently also Eq. (10) and its solution (11), which are used in the following investigations. The authors should comment on this. It becomes also not clear whether the results in Fig. (A3) have been obtained with or without noise.

(7) In the discussion of Fig. 5 on pages 12-13, the authors should comment on the jump in the required laser pulse energy when going from the troposphere to the stratosphere. It is obviously due to the increase of the bin thickness and the resulting larger Rayleigh channel signals. Furthermore the authors should speculate why less energy is required in PBL, compared to the upper troposphere, though the PBL bin thicknesses are smaller, the laser energy and the Rayleigh channel backscattering damping are larger, and ESA's accuracy requirements are more restrictive in PBL. Is there any cross talk from the Mie channel caused by PBL aerosols?

minor specific comments

(8) The authors should be more specific in the abstract in line 15 by writing "increment of averaged Rayleigh channel wind observation uncertainties", since they consider only Rayleigh channel winds.

(9) The authors write in lines 108-109: "Figure 1(b) shows that the solar zenith angle of the observation points of the two new Aeolus-type instruments is low compared to that of Aeolus, and thus lead to larger SBR." However, this figure does not show solar zenith angles. The authors should comment on this.

(10) In lines 273-274: Where do the 18 wind uncertainty profiles come from? And is there 1 profile for every 10° latitude stripe?

(11) Table 2 shows that the averaged increment in the wind observation uncertainties of the 12:00 orbit in the stratosphere is 1.23 m/s, compared to the 18:00 Aeolus orbit. In the text however, 1.4 m/s is reported (lines 16, 286, and 380). Thus the value in the text could be lowered.

(12) The authors should rename the title of Section 4.4 to "Uncertainties of wind observations resulting from an increased laser pulse energy" because they only consider an increased laser pulse energy as a new instrument parameter. Furthermore, the authors should mentioned in line 341 that their proposed laser energy of 80 mJ has been already required by ESA (see e.g. ATBD; Reitebuch et al., 2018). Moreover, the authors should delete the phrase "new instrument parameters, of which" in the caption of Fig. 6. Additionally, the authors should replace "instrument parameters" by "laser energies" in the caption of Tab. 6.

(13) In the abstract, the authors should recall the conditions for which they have derived their results (no clouds, aerosols, noise (?), laser energies of 60 mJ and 80 mJ respectively, number of measurements per observation, number of laser shots per measurement, only Rayleigh channel winds).

The following changes are proposed to improve the readability of the paper.

(14) There are several incidences where different statements are separated only by a comma in one sentence (e.g. lines 10-13). Please check the paper for that and introduce separate sentences.

(15) Please replace "by 0.18, 0.69 m/s" by "by 0.18 and 0.69 m/s" in line 62.

(16) Please provide the reference for the quantum efficiency of the Rayleigh channel detector in line 181 (obviously Reitebuch et al., 2006).

(17) In the caption of Fig. 2, please interchange the 2. and 3. sentence (i.e. first the 3. and then the 2. sentence as the last sentence). Furthermore, do Figs. (c,d) and (e,f) really show numerical differences to Figs. (a,b)? Or do the contours in Figs. (c,d) and (e,f) only show values from the right-hand side scale?

(18) Please add the SBR increments [mW*m^-2*sr^-1*nm^-1] 60.68-20.99=39.69 and 76.36-20.99=55.37 in line 263 because they are listed in the abstract and in the summary.

(19) The sentence in lines 263-264 ("The quantile statistics of SBR is presented in Table 1, which means that the corresponding percentages of the grids (the earth is divided into 1°×1° grid) of which the SBR will be smaller than the values listed in the first line of Table 1.") is unclear. Please provide a clearer formulation, e.g. also by adding an example (e.g., 90% of the grid points (?) or tiles (?) of the 12:00 orbit have SBR values smaller than 105.77 mW*m^-2*sr^-1*nm^-1).

(20) Please replace "upper layer of troposphere and stratosphere" by "upper layer of atmosphere" in lines 277-278.

(21) In the captions of Figs. 3, 5, and 6, the authors write that the "correspondence relationship between the subgraphs and orbits, seasons is consistent with Fig. 2". It is proposed to reformulate this sentence, e.g. to "The arrangement of the subgraphs corresponds to that of Fig. 2".

(22) In lines 298-299: Is the accuracy level of Aeolus, mentioned here, that one shown

in Figs. 3 (a) and (b)? If so, please note this here.

(23) It is assumed that the results shown in Figs. 6 (a) and (b) are identical to those of Figs. 3 (a) and (b). It is however not directly seen due to the different color scales. If so, please make a corresponding note in the text or caption of Fig. 6. If not, please explain why it is not the case.

technical corrections

lines 53-55: Doppler wind lidar which sensing -> senses, Mie/Rayleigh channel sensing -> senses

line 122: expect the mean altitude -> except

Different notations are used for the uncertainty of wind observation in the Rayleigh channel in Eqs. (1) and (8). Please use a consistent notation.

line 233: the wind observation uncertainty which were calculated -> was

line 454: is also need -> needed

line 463: Subsect. 3.4 does not exist, replace by Subsect. 3.3

---

## Author Comment (AC1) · 15 Oct 2020

Dear Dr. Gert-Jan Marseille,

We are truly grateful to your critical comments and thoughtful suggestions. Based on these comments and suggestions, we have made careful thoughts. We are now sending you the corresponding replies. Please point out the mistakes and weaknesses for correction if any. Below you will find our point-by-point responses to your comments/ questions, the comments and suggestions you gave are marked in blue, our replies are

marked in black. Please note that the last figure is Table 1 of the text:

1. The presented results are interesting and potentially useful input for discussions on an Aeolus follow-on mission. The main question, which has not been answered in the paper is: why selecting orbits other than dawn-dusk, given that 3 satellites in a dawn-dusk orbit gives already quite good global coverage, without reducing wind quality due to increased solar background? See Fig.4 of Marseille et al (2008). Is there some indication that different local overpass times would be favorable for NWP? Please elaborate on this in the paper.

**Response:** The main purpose of this manuscript is to access the impact of solar background radiation (SBR) on the accuracy of wind observations for spaceborne Doppler wind lidar (DWL). For spaceborne DWLs operate on sun-synchronous orbits, the dawn-dusk orbit will receive minimum SBR, and the noon orbit will receive maximum SBR. The spaceborne DWLs operate on the sun-synchronous orbits with local time of ascending node (LTAN) crossing of 15:00 will receive medium SBR.

Three spaceborne DWLs operate on a dawn-dusk orbit give quite good global coverage, however, they would receive similar SBR. To access the impact of orbits selection on the accuracy of wind observations, the three orbits with LTANs of 18:00, 15:00, and 12:00 were purposed. The influence of spaceborne DWLs operate on different orbits on NWP was not considered in this paper, because this is not the focus of this manuscript, but it is a very interesting topic that deserves further study.

2. In connection to this. Line 43: "The future spaceborne DWLs may operate on different orbits which should be related to their observation purposes". Which observation purposes are related to 12:00 or 15:00 local overpass times? See also line 72: "Assuming the future Aeolus-type spaceborne DWLs will operate on the sun-synchronous orbits with different LTAN". Based on what assumption?

**Response:** Apart from Aeolus, the hybrid Doppler wind lidar (HDWL) designed by US would utilize both direct-detection and coherent-detection technology to observe verti-

cal profile horizontal LOS (line-of-sight) wind vectors. SBR would lower the accuracy of wind observed by direct-detection technology. According to the National Polar-orbiting Operating Environmental Satellite System (NPOESS) mission, US would launch two HDWLs in two stages. Stage 1: Global Wind Observing Sounder (GWOS) mission: the mission aims to demonstrate the prototype HDWL system whether it would be capable of global wind measurements. GWOS would operate on a low earth orbit with orbit height of 400 km, the detailed parameters of the orbit are unknown. Stage 2: NPOESS Wind Observing Sounder (NWOS) mission: the mission would launch a HDWL system carried on the NPOESS satellites that would meet fully-operational wind measurement requirements. The NPOESS satellite constellation consists of 3 satellites in 828 km altitude, sun synchronous orbits, with LTANs of 13:30, 17:30, and 21:30. It is not clear which orbit the NWOS will operate. According to the stowed configuration of three satellites and the location of the NWOS shown in Figure 1 and 2, it is inferred that NWOS may operate on the orbit of 21:30.

This satellite orbits were designed in 15 year ago, and may be adjusted before the launch of satellites in the future. However, the fact illustrates that the future spaceborne DWL may operate not only on the sun synchronous dawn-dusk orbit. To some extent, the research in this manuscript will also provide references for the orbit selection of HDWL in NWOS.

On the other hand, supposed that one satellite constellation consists three spaceborne DWLs operate on sun synchronous orbits with LTANs of 18:00, 15:00, and 12:00, we can reconstruct the wind speed diurnal cycle according to the wind observations. For both wind observations and analyzed wind acquired from NWP shows the obvious diurnal characteristics of wind field: 1) the diurnal variations of wind speed follow cosine curve, approximately; 2) wind speed reaches its maximum value at about 23:00; 3) the analyzed wind speed has peak value during the sunrise (Zhang and Zheng, 2004, Holtslag et al., 2013).

3. The realism of the simulations can be largely improved by comparing Aeolus measured SBR with simulated Aeolus SBR. Information on Aeolus measured SBR and the impact on Aeolus wind quality is found in the attached supplementary material. Based on this information, the authors can test the realism of their simulations.

**Response:** The solar background noise (SBN) received by Aeolus is determined by the geometry of solar and satellite, atmospheric conditions, and earth reflectance. And the solar zenith angle (SZA) of the off-nadir points is the main determinants. The simulated SZAs of the off-nadir points within one-year range are shown in Fig. 3. When the solar zenith angle is greater than 90 degrees as the horizontal red line shows, the received SBR could be negligible. Comparisons between Fig. 3 and Fig. 1 of the supplement provided by reviewer 1 show high consistence. Both of them are periodic. The values of them reach maximum near summer solstice and reach maximal near winter solstice. And the values of SBR reach minimal values near spring and autumn equinox. The 4 time ranges in Fig. 3 divided by 8 red lines denote 15 days near autumn equinox, winter solstice, spring equinox, summer solstice.

It requires great amount of computing to simulate the received SBR of Aeolus with one-year range. The received SBR within 15 days near summer and winter solstice was simulated. And the SBR in summer and winter solstice was converted to ACCD counts of Rayleigh channel as is illustrated in Fig. 4.

As Fig. 4 illustrated, the amount of ACCD counts near summer and winter solstice are consistent with Fig. 1 of the supplement. And solar background noise excited on Rayleigh channel are periodic as the subgraph of the supplement shows.

In order to facilitate reviewers to verify the simulation model of Aeolus used in this manuscript, we would introduce the method and the parameters used in this manuscript which are mainly derived from ATBD ADM-Aeolus Level1B Products (issue 3.0, 30.11.2006) (Reitebuch et al., 2006). The energy of solar background noise denotes in photon counts before Rayleigh channel can be expressed as

$$n_{solar} = n_{meas} \cdot n_{pulse} \cdot C(\lambda) \cdot L_\lambda(\Theta, \psi, \lambda) \cdot \Omega_0 \cdot A_0 \cdot t_D \cdot \Delta\lambda \cdot \lambda / (h \cdot c), \qquad (1)$$

where $n_{meas}$ is the number of measurements in one observation. $n_{pulse}$ is the number of pulses in one measurement. $C(\lambda)$ denote the transmission of optical instrument of ALADIN. $L_\lambda(\Theta, \psi, \lambda)$ denote the solar background radiation. $\Omega_0$ denote the acceptance solid angle. $A_0$ is the area of telescope. $t_D$ is the detection time for the solar background range gate. $\Delta\lambda$ is the bandwidth of the receiver. $\lambda$ denote the wavelength of the laser. And $h$ denote Plank constant, $c$ denote the light speed.

The energy excited by SBN before the Rayleigh channel can be computed using Eq. (1). Assuming that the spectrum of SBN follows uniform distribution, and the energy is equal to the result of Eq. (1), its bandwidth equal to the Free Spectral Range of Rayleigh channel, the photon counts excited on Rayleigh channel can be obtained after transmitting through the Fabry-Perot interferometer and multiplying by the quantum efficiency of ACCD.

According to the latest issue of Aeolus ATBD L1B Products (Reitebuch et al., 2018), the latest parameters of ALADIN is different from the parameters we used in the manuscript, as is mentioned by reviewer 2. The comparisons of the parameters are illustrated in Table 1 as the last figure shows at the end of the text.

The simulated SBN using old parameters is shown in Fig. 5. The comparison between Fig. 4 and 5 illustrated that larger energy is excited using old parameters. More laser pulses are accumulated in one observation and the wider bandwidth of the FWHM and Free Spectral Range of FP interferometer in old parameters can account for this phenomenon.

Then, under the same atmospheric conditions, how much difference will the wind observations uncertainties obtained by using the new and old parameters be? The simulated uncertainties of wind observations under cloud-free atmospheric condition are illustrated in Fig. 6. The corresponding solar zenith angle of Fig. 6 is $70°$, and the related solar background radiation is 72.19 mW$\cdot m^{-2}\cdot sr^{-1}\cdot nm^{-1}$. As Fig. 6 (a) shows, the difference of uncertainties simulated using old and new parameters is large. The

largest and average difference is 2.17 and 0.61 m/s, respectively. Fig. 6 (b) illustrated that the SNR simulated using new parameters is obviously large than the results using old parameters. The combination of Figs. 6 (c, d) can account this phenomenon. In Fig. 6 (c), the simulated useful signal of Rayleigh channel is relatively close, almost no difference. However, Fig. 6 (d) illustrated that the simulated solar background noise obtained using old parameters is much large than that of new parameters. Why the solar background noise simulated using old parameters is much larger than that of new parameters? As Tab. 1 illustrated, under old parameters, the Free Spectral Range of Rayleigh channel and FWHM of the FP interferometers of the old parameters is much larger than that of the new parameters. And in the simulation process, the SBN is regarded as following uniform distribution. Wider transmission bandwidth of Rayleigh channel will lead to higher solar background energy, which would lower the SNR of Rayleigh channel, and then increase the uncertainties of wind observations.

To verify the correctness of our simulation model, we reconstructed Fig. 2 of the supplement using new parameters of Tab. 1 and the results are shown in Fig. 7. In the simulation, the typical and worst solar background radiation are set as 72.50 and 156.00 mW$\cdot m^{-2} \cdot sr^{-1} \cdot nm^{-1}$. The comparisons between Fig. 7 and Fig. 2 of the supplement show large difference. In Fig. 2 of the supplement, when the useful signal in channel A reach 5000, the related wind observation uncertainty is about 4 m/s. In our simulation, the uncertainty is about 8 m/s when the useful signal in channel A is about 5000. However, the uncertainties of wind observation are about 2~3 m/s when the solar background radiation is about 72.19 mW$\cdot m^{-2} \cdot sr^{-1} \cdot nm^{-1}$. The results are reasonable. In our simulation, the photon counts excited by typical and worst solar background noise in channel A are 1.34*10$^4$ and 2.92*10$^4$ respectively when the vertical height of the range gate is 1 km. In Fig. 6, the useful signal in channel A is general between 2*10$^4 \sim$ 3*10$^4$. I suppose the cause for this phenomenon maybe different instrument parameters used in our simulation or the polarization effects may not considered in our simulation, which lead to lager simulation results of the SBN and useful signal of channel A. The Matlab script used to simulate the results of Fig. 7 are

attached in the supplement for further discussion. Please point out and give directions to us whenever you see any weaknesses or shortages within.

===========Minor comments ============

4. line 60: "The received SBR of Aeolus ranges from 0 to 169 mW·m-2·sr-1·nm-1". That is worse than "On the two new orbits, the increments of averaged SBR received by the new spaceborne DWLs range from 39 to 56 mW·m-2·sr-1·nm-1" (line 14). Can the author please comment.

**Response:** Line 14: "On the two new orbits, the increments of averaged SBR received by the new spaceborne DWLs range from 39 to 56 mW·$m^{-2}$·$sr^{-1}$·$nm^{-1}$." The sentence is referred to that the averaged SBR received by Aeolus-type instruments operate on the sun-synchronous orbits with LTANs of 15:00 and 12:00 is higher than Aeolus, and the average increment of SBR ranges from 39 to 56 mW·$m^{-2}$·$sr^{-1}$·$nm^{-1}$. The conclusion is corresponding to the sentence: "Statistics illustrate that the averaged SBR of the three spaceborne DWLs are 20.99, 60.68, and 76.36 mW·$m^{-2}$·$sr^{-1}$·$nm^{-1}$ respectively." (line 262).

The sentence in line 14 may be ambiguous. In the revision, line 262 will be modified to make the sentence in line more clear as reviewer 2 suggest.

**Modified:** Statistics illustrate that the averaged SBR of the three spaceborne DWLs are 20.99, 60.68, and 76.36 mW·$m^{-2}$·$sr^{-1}$·$nm^{-1}$ respectively. The increments of averaged SBR received by the new spaceborne DWLs are 60.68-20.99=39.69 mW·$m^{-2}$·$sr^{-1}$·$nm^{-1}$ and 76.36-20.99=55.37 mW·$m^{-2}$·$sr^{-1}$·$nm^{-1}$.

5. Figure 1b is misleading since it suggests that all three Aeolus-type instruments operate during daytime at equal time intervals.

**Response:** Figure 1b illustrates the off-nadir points on earth surface of the three spaceborne DWLs operate on the sun-synchronous orbits with LTANs of 18:00, 15:00, and 12:00, respectively.

6. line 108: "Figure 1(b) shows that the solar zenith angle of the observation points of the two new Aeolus-type instruments is low compared to that of Aeolus". How can that be seen from the figure?

**Response:** In Figure 1 (b), the shaded area represents the night, the non-shaded area represents the day, and the dividing line between the shaded area and the non-shaded area is the dividing line between day and night. In Figure 1b, the trajectory of the off-nadir points of Aeolus is closer to the dividing line compared to the off-nadir points of Aeolus2 and Aeolus3. Therefore, it is supposed that the solar zenith angles of the off-nadir points of the two new orbits should be lower.

As the referee pointed, it is not clear to see the comparative relationship in the sun zenith angle of the three orbits. In the latter revision, we plan to add a figure to illustrate the fact in the revision, as Fig. 8 shows, which illustrate the variations of solar zenith angle of the three orbits as time.

7. The simulations would be more useful if the operational Aeolus instrument would be used for reference 1. one measurement is composed of 20 accumulated shots onboard; 1 observation is obtained from averaging 30 measurements. 2. Assuming around 60mJ laser energy, which is consistent with current operational Aeolus laser-B 3. optical throughput is a factor 2-3 lower than expected. The authors can do simulations based on both this unexpected signal loss (worst case scenario) and without this loss, assuming that the problem can be identified and solved before the launch of the Aeolus follow-on mission (best case scenario). With the above settings 0% of Aeolus data would meet the mission requirement, rather than 88.01% as mentioned in Table 5. So, it would be interesting to extend Table 5, by presenting both best and worst case scenarios.

**Response:** In the manuscript, Aeolus was assumed to be operated on best case scenario. As is illustrated in Fig. 6, Aeolus performs better using new parameters. It is very meaningful if we can assess the impact of instrument operational instrument on

the wind observation uncertainties combined with the operational Aeolus instrument. This is also a topic that I'm very interested in. I think it can be studied as the next topic when we could get access to Aeolus L0 measurement product.

8. line 118: "we focus on the simulation of the wind retrieved method on Rayleigh channel, and assume that the cross-talk effect between Mie channel and Rayleigh channel is negligible". Based on this assumption, you can remove mentioning over scattering ratio in section 3.1.

**Response:** Thanks for your kind mention. In the revision, we will remove related expressions.

9. Derivation of Eq. (7) in Appendix could have been done more simple, by substituting A=B in Eq.(3) => $\sigma_{R_{ATM}} = \sigma_A/(N_A * sqrt(2))$ Substituting Eq. (4) in Eq.(6) and setting A=B => $SNR_{Ray} = (N_A * sqrt(2))/\sigma_A$

**Response:** Thanks for your kind suggestion. the method would make the derivation much more simple which should also assumed that $N_{S,A} = N_{S,B}$. In revisions, we will simplify the derivation method according to your suggestions.

10. Figure 2. For the 15:00 and 12:00 UTC orbits, half the orbit is in full darkness (so no SBR contribution), the other half in full daylight (a large SBR contribution). How is this reflected in figure 2?

11. Also in Figure 3, I would expect bi-modal accuracy statistics with very good quality at the dark part of the orbit (no SBR) and low quality in the day-light part (high SBR). So, what is exactly displayed in Figure 3? Please present both statistics separately. In the caption of figure 3, mention that this is winds from the Rayleigh channel in clean air conditions.

**Response:** Here we will respond to the above two questions together.

The main topic of this manuscript is to discuss the impact of SBR on the wind observation accuracy of spaceborne DWLs on different orbits, and how much the laser energy

should be set to achieve required accuracy. Therefore, we put emphasis on the worst cases of SBR.

In Fig. 2 of the manuscript, the maximum SBR of each grid (the earth was divided into $1° \times 1°$ grids) was illustrated. Because the SBR is not much different at the same latitude as Fig. 2 of the manuscript shows, the SBR are averaged within $10°$ latitude. Then the $10°$ latitude averaged atmospheric conditions were obtained from Ozone Monitoring Instrument (OMI) database as mentioned in subsection 2.2 of the manuscript. Finally, the $10°$ latitude averaged uncertainties of wind observation on Rayleigh channel derived and show in Fig. 3 of the manuscript. Therefore, the Fig. 2 and Fig. 3 of the manuscript are the worst cases of SBR with maximum Rayleigh channel uncertainties as the reviewer 2 suggests. The detailed reasons were described as follows.

Fig. 3 and Fig .1 of the supplement illustrate that the determinant of SBR for Aeolus is SZA. As Fig. 3 shows, the values of SBR received by Aeolus will reach maximum and maximal values near summer and winter solstice. To derive the largest SBR, the time zones near summer and winter solstice were selected.

It is found that we only briefly described how to calculate the SBR for single off-nadir points in the manuscript as the last paragraph of section 2.2 of the manuscript shows, but didn't not describe how the global distributions of SBR were obtained. In the revision, we will add relevant descriptions. The details can also refer to subsection 3.1 of (Zhang et al., 2019), and was described briefly here. After we derived the SBR of each off-nadir point, earth is divided into $1° \times 1°$ grids. Each $1° \times 1°$ grid would include several off-nadir points, and the maximum TOA radiance in the grid was picked as the value of SBR in this grid. Finally we could derive the global distributions of SBR using the grids. And the global distributions of SBR during 15 days near summer and winter solstice are illustrated in Fig. 2 of the manuscript.

As is indicated in Fig. 8, the received SBR reach maximum value near summer solstice for Aeolus as a whole. For each grid of the earth, does the values of SBR reach

maximum values near summer and winter solstice? The distribution of SZAs of 5 grids within one-year range of the three orbits are illustrated in Fig. 9. Fig. 9 illustrates that the SZAs of off-nadir points in North Hemisphere reach maximum values near summer solstice, and the SZAes of off-nadir points in Sorth Hemisphere reach maximum values near summer solstice. However, for the off-nadir points near equator, the SBR reach minimal value. Consider from majority off-nadir points, the time zones near summer and winter solstice were taken as the worst cases.

Because we did not explain why we select the SBR near summer and winter solstice to analyse in this manuscript, it may cause confusion for readers. In the revision, we will add the related expressions.

**References**

HOLTSLAG, A., SVENSSON, G., BAAS, P., BASU, S., BEARE, B., BELJAARS, A., BOSVELD, F. C., CUXART, J., LINDVALL, J., STEENEVELD, G. J., TJERNSTROM, M. Van de WIEL, B. (2013), "STABLE ATMOSPHERIC BOUNDARY LAYERS AND DIURNAL CYCLES Challenges for Weather and Climate Models", BULLETIN OF THE AMERICAN METEOROLOGICAL SOCIETY, Vol. 94 No. 11, pp. 1691-1706.

NAEGELI, C., SJOBERG, B., SCHNEIDER, S., LEE, P., FARA, D., ADKINS, D. ANDREOLI, L. (2004) National Polar-orbiting Operational Environmental Satellite System (NPOESS) Potential Pre-planned Product Improvement (P3I) Status. AGU Fall Meeting. REITEBUCH, O., HUBER, D. NIKOLAUS, I. (2018) ATBD: ADM-Aeolus Level 1B Product., ESA.

REITEBUCH, O., PAFFRATH, U. LEIKE, I. (2006) ATBD: ADM-Aeolus Level 1B Product., ESA.

ZHANG, C., SUN, X., ZHANG, R., ZHAO, S., LU, W., LIU, Y. FAN, Z. (2019), "Impact of solar background radiation on the accuracy of wind observations of spaceborne Doppler

wind lidars based on their orbits and optical parameters", Optics Express, Vol. 27 No. 12, pp. A936-A952.

ZHANG, D. L. ZHENG, W. Z. (2004), "Diurnal cycles of surface winds and temperatures as simulated by five boundary layer parameterizations", JOURNAL OF APPLIED METEOROLOGY, Vol. 43 No. 1, pp. 157-169.

Please also note the supplement to this comment:
https://amt.copernicus.org/preprints/amt-2020-202/amt-2020-202-AC1-supplement.zip

![Stowed configuration diagram of NPOESS satellite with labeled components: HRD ANTENNA, APS, S-BAND (NADIR), SARSAT-Tx, SESS HORUS, VIIRS, CMIS, SMD ANTENNA, SURVIVABILITY, SAR/ADCS-Rx, LRD ANTENNA. Measurements shown: 0.60m² 133kg, 0.43m² 72kg, 0.56m² 90kg, 0.19m² 47kg, 1.0m² 72kg. Axes labeled +Y_S/C and +X_S/C. Legend: Unused Real Estate 0.25m X 0.25m. STOWED CONFIGURATION]

**Fig. 1.** Stowed configuration of NPOESS satellite operates on the orbit of 21:30 (Naegeli et al., 2004).

**Fig. 2.** The location of HDWL of NWOS in NPOESS satellite. (by DWL Mission Definition Team, 2005).

**Fig. 3.** The simulated solar zenith angle of the off-nadir points of Aeolus within one-year range.

[Figure]

**Fig. 4.** The received solar background noise of Rayleigh channel (A+B) within 1 day. (a) summer solstice, (b) winter solstice.

[Figure]

**Fig. 5.** Same as Fig. 4 using old instrument parameters shown in Tab. 1.

[Figure]

**Fig. 6.** Some simulation results obtained using old and new instrument parameters. (a)wind observations uncertainties; (b)Signal to noise ratio; (c)Useful signal; (d)Signal excited by SBN.

[Figure]

Fig. 7. The relationship between uncertainties of wind observations and useful signal of channel A on Rayleigh channel.

[Figure]

**Fig. 8.** The variations of SZA of the off-nadir points on the three orbits within one-year range. Sun-synchronous orbit with Local Time of Ascending Node crossing (LTAN) of 18:00 (a), 15:00(b) and 12:00(c).

[Figure]

**Fig. 9.** Distributions of SZAs of 5 grids of three orbits within one-year range. Coordinates are (60°,0°), (30°,0°), (0°,0°), (-30°,0°), and (-60°,0°). Three rows are the 18:00, 15:00, and 12:00 UTC orbits.

| Unit | Parameter | Symbol | Value | |
|---|---|---|---|---|
| | | | **New** | **Old** |
| ALADIN instrument | Number of measurements per observation | $n_{meas}$ | 30 | 14 |
| | Number of pulses per measurement | $n_{pulse}$ | 20 | 50 |
| | Optical transmission of instrument | $C(\lambda)$ | 0.34 | 0.8 |
| | Field of View (FOV) | $\theta$ | 18.1 μrad | 22 μrad |
| | Diameter of telescope | $D$ | 1.5 m | 1.5 m |
| | Vertical length of 25th range gate | $\Delta z$ | 200 km | 200 km |
| | Bandwidth of the receiver | $\Delta\lambda$ | 1 nm | 1 nm |
| | Wavelength of the receiver | $\lambda$ | 355 nm | 355 nm |
| | Plank constant | $h$ | $6.626*10^{-34}$ | $6.626*10^{-34}$ |
| | Light speed | $c$ | $3*10^8$ | $3*10^8$ |
| Rayleigh Spectrometer | Fabry-Perot Free Spectral Range | | 4.56 pm | 4.6 pm |
| | Peak transmission | | 81% / 67% | 36.8%/27.2% |
| | Filter separation | | 2.33 pm | 2.65 pm |
| | Filter FWHM (direct/reflected) | | 0.65pm/ 0.64 pm | 0.74 pm / 0.70 pm |
| | Quantum efficiency | | 84% | 75% |

**Fig. 10.** Table 1. Main ALADIN instrument parameters.

---

## Author Comment (AC2) · 15 Oct 2020

Dear Dr. Karsten Schmidt,

We are truly grateful to your critical comments and thoughtful suggestions. Based on these comments and suggestions, we have made careful thoughts. We are now sending you the corresponding replies. Please point out the mistakes and weaknesses for correction if any. Below you will find our point-by-point responses to your comments/ questions, the comments and suggestions you gave are marked in blue, our replies are

marked in black:

Major specific comments

1. The authors should compare SBR computed by means of their model with measured in-orbit SBR data for certain time ranges. The reviewer 1 provided some data in his re- view supplement. This would increase the confidence in the authors model. Moreover, plots over a year show that SBR measured by Aeolus is maximum in June and De- cember (see again supplement). Thus the authors can argue that their investigations for June and December are for the worst cases with maximum Rayleigh channel wind errors due to SBR.

**Response:** Thanks for your suggestions. As to the comparison between simulated and measured solar background radiation (SBR), we explained the topic in detail in the reply to Q3 of reviewer 1. The main ideas are described briefly as follows:

First, because the SBR is mainly determined by solar zenith angle of the off-nadir points, we computed the solar zenith angles of off-nadir points within one-year range. The variations of solar zenith angles are in consistent with the variations of in-orbit measured SBR indicate that the variation trend of simulated SBR would be in consis- tent with the measured SBR.

Then, we simulated SBR received by Aeolus during 15 days near summer and winter solstices. And the simulated SBR in the two days of summer and winter solstices was shown in Fig. 4 of the reply to reviewer 1 to compare with Fig. 1(a) of the supplement. The comparisons show that the two are consistent in variation trend and magnitude, which increase the confidence of our model.

In addition, in the ATBD L1B Products (issue 4.4 20.04.2018), the highest background integration times is 3750 $\mu$s which is related to 446 km vertical height of 25th range gate. In our simulation, the integration times for solar background radiation is 1680 $\mu$s which is related to 200 km vertical height of 25th range gate.

Furthermore, thanks very much for your suggestions in reply to reviewer 1, we have stated in the response to reviewer 1 that we are considering the worst conditions of solar background radiation in the manuscript.

2. In lines 60-61, the authors write that the "received SBR of Aeolus ranges from 0 to 169 mW*m-2*sr-1*nm-1". The authors should give the corresponding reference. Aeolus measures primarily ACCD counts of SBR.

**Response:** Thanks for your kind reminder. The results were obtained from (Zhang et al., 2019). We will add the citation in the revision.

3. It is of course possible and interesting to consider sun-synchronous orbits other than dawn-dusk orbits. The authors should explain their choice of orbits with LTANs of 15:00 and 12:00 in Section 2.1.

**Response:** Thanks for your reminder. We will add the related explanations in Section 2.1 in the revision. The detailed reasons were given in Q1 of the reply to reviewer 1.

4. It becomes not clear which kinds of aerosols are considered by the authors in their simulations (only aerosols in the planetary boundary layer (PBL) or also above it). The authors should specify this. Furthermore, the authors should replace "clear sky" by "cloud-free" in lines 15 and 379 due to the presence of aerosols.

**Response:** In the manuscript, only the aerosols in the PBL were considered. The aerosol above it (stratospheric aerosols) were not taken into account.

The optical properties of aerosols used in this manuscript was obtained from the LIVAS (LIdar climatology of Vertical Aerosol Structure for space-based lidar simulation studies) database (Amiridis et al., 2015). In the database, the products of aerosol optical properties include "355_Aerosol_Backscatter_Mean", "355_Stratospheric_Backscatter_Mean", and "355_Total_Backscatter_Mean". "355_Aerosol_Backscatter_Mean" was used as the backscatter coefficients of aerosols in this manuscript.

In the revision, we will replace "clear-sky" by "cloud-free" in lines 15 and 379. Thanks for your detailed suggestions.

5. For the simulations, the Aeolus instrument parameters have been taken by the authors from the Algorithm Theoretical Basis Document (ATBD; Reitebuch et al., 2006). There is however a newer version of this document (issue 4.4, 20.04.2018), e.g. available by ESA for Aeolus CalVal users. Furthermore, the authors considered observations consisting of 50 accumulations (measurements) of 14 shots, resulting in a horizontal resolution of about 100.8 km per observation. However, Aeolus has 30 measurements per observation with 20 laser pulses per measurement (in the level 1B processing), resulting in a horizontal averaging length of about 90km per observation. So the averaged wind observation uncertainties, derived by the authors in the present study, are only some estimates. It is proposed to use the newer/current parameters in future simulations in order to increase their usefulness.

**Response:** Thanks very much for your reminder. Tab. 1 (Fig. 10 at the end of the text) of the reply to reviewer 1 illustrate that the parameters used in our simulation and the new parameters are of quite difference, which are mainly reflected in narrower bandwidth of Fabry-Perot Free Spectral Range and FWHM of Rayleigh channel of the new instrument parameters. Under new instrument parameters, the same values of SBR would excite fewer photon counts on the ACCD of Rayleigh channel. However, the photon counts excited by atmospheric backscattered signals are similar under new and old parameters. Finally, the fact would lead to smaller wind observation uncertainties using new parameters.

Our simulation indicates that the largest and mean difference of wind observation uncertainties were 2.17 and 0.61 m/s under a specific atmospheric condition and the SBR of 72.19 mW$\cdot m^{-2} \cdot sr^{-1} \cdot nm^{-1}$. The details of the experiment can refer to Q3 of the reviewer 1.

Given the large difference between the old and new parameters, if the associate editor

give the chance to revise, we will use the new parameters in our simulation model.

6. Eq. (7) has been numerically verified in the Appendix by neglecting noise (see item (5) in line 436). Consequently, Eq. (8) holds only for this restriction. Then, Eq. (8) is reformulated to Eq. (10) by using Eq. (6). However, Eq. (6) does contain noise, and consequently also Eq. (10) and its solution (11), which are used in the following investigations. The authors should comment on this. It becomes also not clear whether the results in Fig. (A3) have been obtained with or without noise.

**Response:** The results in Fig. (A2, A3) have been obtained without noise.

When deriving the formula, we considered the influence of the noise of the detection unit on the accuracy of wind observations, so all formula derivations include noise, including Eq. (7). As Tab. 4-1 of ADM-Aeolus ATBD Level1B Products shows, the detection chain noise for each measurement is 4.7 e-/pixel, and the dark current is 1.9 e-/(pixel·s), which is negligible compared to the photon counts excited by atmospheric backscattered signal and solar background radiation. Therefore, in the verification for the equations in Appendix (Fig. A2 and A3), the noise is not taken into account.

Is it reasonable to neglect the noise in verification the noise in the simulation? If not, we will taken the noise into account in the verification. If so, we will explain the reason why we neglect the noise in Appendix.

7. In the discussion of Fig. 5 on pages 12-13, the authors should comment on the jump in the required laser pulse energy when going from the troposphere to the stratosphere. It is obviously due to the increase of the bin thickness and the resulting larger Rayleigh channel signals. Furthermore the authors should speculate why less energy is required in PBL, compared to the upper troposphere, though the PBL bin thicknesses are smaller, the laser energy and the Rayleigh channel backscattering damping are larger, and ESA's accuracy requirements are more restrictive in PBL. Is there any cross talk from the Mie channel caused by PBL aerosols?

**Response:** Thanks for your kind reminder, we will explain the jump when going from troposphere to stratosphere in line 321 of Page 12.

**Original text:** Higher energy is needed mostly in the upper level of troposphere and stratosphere near the regions close to Antarctic and Arctic circles. The closer the orbital LTAN is to noon, the averaged values of the required laser energy will become larger.

**Modified:** Higher energy is needed mostly in the upper level of troposphere and stratosphere near the regions close to Antarctic and Arctic circles. On the boundary line with a height of 16 km, there is an obvious sudden jump in required laser energy, and the required laser energy is reduced. This is mainly because the vertical thickness of measurement bins changes from 1 km to 2 km at height of 16 km, which makes the integration time of detection units of Rayleigh channel double. And larger atmospheric backscattered signal would be received. On the other hand, the required wind observation uncertainty increase from 2 m/s to 3 m/s in the stratosphere. Therefore, the required laser energy reduced suddenly when going from troposphere to stratosphere. The comparisons among the required laser energy of the three orbit illustrate that the closer the orbital LTAN is to noon, the averaged values of the required laser energy will become larger.

*Solar background noise has main impact on the wind observation uncertainties on Rayleigh channel. The impact of SBR on Mie channel is negligible (Rennie, 2017). Due to the widespread presence of aerosols in PBL, Mie channel is used to observe the wind in PBL. And the main topic of the manuscript is to study the impact of SBR on the Rayleigh channel wind observations. Therefore, the Rayleigh wind observations in PBL is not considered. Sentences in the original text may misunderstand readers, we will modify as follows:*

**Original text:** In fact, the Mie channel is mostly used for wind observations in the PBL, which are of higher accuracy. It is meaningless to study the wind observation accuracy

of the Rayleigh channel in the PBL, the accuracy of the Rayleigh channel in the PBL is not considered in the following of this paper. (line 279)

**Modified:** In fact, the Mie channel is mostly used for wind observations due to the widespread presence of aerosols in PBL. Therefore, the accuracy of the Rayleigh channel in the PBL is not considered in the following of this paper.

*As to the questions why less energy is required in PBL, in the PBL, the Mie channel wind observation uncertainties is much less than that of Rayleigh channel as Fig. 1(a) at the end of the text shown. However, the photon counts of Mie channel excited by atmosphere aerosols and solar background radiation is also much less than that of Rayleigh channel as Fig. 1(b, c) shown. The optical properties of aerosols are obtained from RMA dataset. So I don't know the reasons for the phenomenon that the wind observation accuracy on Mie channel is higher than that of Rayleigh channel. It may be due to the detection mechanism of Mie channel. Hope for further discussions to the question with reviewers.*

minor specific comments

8. The authors should be more specific in the abstract in line 15 by writing "increment of averaged Rayleigh channel wind observation uncertainties", since they consider only Rayleigh channel winds.

**Response:** Thanks very much for your suggestions. We will modify the text according to your suggestions in the revision.

9. The authors write in lines 108-109: "Figure 1(b) shows that the solar zenith angle of the observation points of the two new Aeolus-type instruments is low compared to that of Aeolus, and thus lead to larger SBR." However, this figure does not show solar zenith angles. The authors should comment on this.

**Response:** Thanks for your reminder. From Fig. 1(b), we cannot get the information that the solar zenith angle of two new Aeolus-type instruments is low compared to that

of Aeolus. We will add a graph to illustrate the point, as Fig. 9 of the reply to reviewer 1 shows.

10. In lines 273-274: Where do the 18 wind uncertainty profiles come from? And is there 1 profile for every 10? latitude stripe?

**Response:** Yes, there is 1 profile for every $10°$ latitude stripe.

In Fig. 2 of the manuscript, the maximum SBR of each grid (the earth was divided into $1°×1°$ grids) was illustrated. Because the SBR is not much different at the same latitude as Fig. 2 of the manuscript shows, the SBR are averaged within $10°$ latitude. Then the $10°$ latitude averaged atmospheric conditions were obtained from Ozone Monitoring Instrument (OMI) database as mentioned in subsection 2.2 of the manuscript. Finally, the $10°$ latitude averaged uncertainties of wind observation on Rayleigh channel derived and show in Fig. 3 of the manuscript.

11. Table 2 shows that the averaged increment in the wind observation uncertainties of the 12:00 orbit in the stratosphere is 1.23 m/s, compared to the 18:00 Aeolus orbit. In the text however, 1.4 m/s is reported (lines 16, 286, and 380). Thus the value in the text could be lowered.

**Response:** Thanks very much for kind reminder, we will lower the value in the corresponding text.

12. the authors should rename the title of Section 4.4 to "Uncertainties of wind observations resulting from an increased laser pulse energy" because they only consider an increased laser pulse energy as a new instrument parameter. Furthermore, the authors should mentioned in line 341 that their proposed laser energy of 80 mJ has been already required by ESA (see e.g. ATBD; Reitebuch et al., 2018). Moreover, the authors should delete the phrase "new instrument parameters, of which" in the caption of Fig. 6. Additionally, the authors should replace "instrument parameters" by "laser energies" in the caption of Tab. 6.

**Response:** Thanks for your kind suggestions.

We will rename the title of Section 4.4 to "Uncertainties of wind observations resulting from an increased laser pulse energy".

The sentence in line 341 will be replaced with "while taking the existing technical level into account, the laser energy of the two new spaceborne DWLs is set to 80 mJ, which has been already required by ESA for Aeolus." And add the corresponding citation.

The caption of Fig. 6 will be renamed as "The zonal distributions of wind observation uncertainties of the three spaceborne DWLs, the laser energy of Aeolus is 60 mJ, and the laser energy of the two new Aeolus-type spaceborne DWLs is 80 mJ."

The caption of Tab. 6 will be renamed as "The averaged wind observation uncertainties of the three spaceborne DWLs with the proposed laser energies."

13. In the abstract, the authors should recall the conditions for which they have derived their results (no clouds, aerosols, noise (?), laser energies of 60 mJ and 80 mJ respectively, number of measurements per observation, number of laser shots per measurement, only Rayleigh channel winds).

**Response:** Thanks for reminder. We modified the original text as follows:

**Original text:** the impact of the local time of ascending node (LTAN) crossing of sun-synchronous orbits on the wind observation accuracy was studied in this paper by proposing two added Aeolus-type spaceborne DWLs operated on the sun-synchronous orbits with LTAN of 15:00 and 12:00 combined with Aeolus. (line 11)

**Modified:** the impact of the local time of ascending node (LTAN) crossing of sun-synchronous orbits on the Rayleigh channel wind observation accuracy was studied in this paper by proposing two added Aeolus-type spaceborne DWLs operated on the sun-synchronous orbits with LTAN of 15:00 and 12:00 combined with Aeolus.

**Original text:** On the two new orbits, the increments of averaged SBR received by the

new spaceborne DWLs range from 39 to 56 mW·$m^{-2}$·$sr^{-1}$·$nm^{-1}$ under clear skies, which will lead to the increment of averaged wind observation uncertainties from 0.3 to 0.4 m/s in the troposphere and from 0.9 to 1.4 m/s in the stratosphere. (line 13)

**Modified:** On the two new orbits, the increments of averaged SBR received by the new spaceborne DWLs range from 39 to 56 mW·$m^{-2}$·$sr^{-1}$·$nm^{-1}$ under cloud-free conditions, which will lead to the increment of averaged wind observation uncertainties from 0.3 to 0.4 m/s in the troposphere and from 0.9 to 1.4 m/s in the stratosphere on Rayleigh channel. In our simulation, one observation consists of 14 measurements and 50 laser pulses are accumulated in one measurement.

The following changes are proposed to improve the readability of the paper.

14. There are several incidences where different statements are separated only by a comma in one sentence (e.g. lines 10-13). Please check the paper for that and introduce separate sentences.

**Response:** Thanks for your suggestions. We will make corresponding modifications in the revision.

15. Please replace "by 0.18, 0.69 m/s" by "by 0.18 and 0.69 m/s" in line 62.

**Response:** Thanks for your suggestions. We will make corresponding modifications in the revision.

16. Please provide the reference for the quantum efficiency of the Rayleigh channel detector in line 181 (obviously Reitebuch et al., 2006).

**Response:** Thanks for your suggestions. We will provide the reference value for the quantum efficiency and add related citation.

17. In the caption of Fig. 2, please interchange the 2. and 3. sentence (i.e. first the 3. and then the 2. sentence as the last sentence). Furthermore, do Figs. (c,d) and (e,f) really show numerical differences to Figs. (a,b)? Or do the contours in Figs. (c,d) and

(e,f) only show values from the right-hand side scale?

**Response:** Thanks for your suggestions. We will interchange the 2. and 3. sentence of the caption of Fig. 2. The caption will be modified to "Global distributions of SBR received by spaceborne DWLs operated on the three orbits. Figs. (a, b), (c, d) and (e, f) present the sun-synchronous orbits with LTAN of 18:00, 15:00, and 12:00 respectively, and the upper panels denote the SBR in summer, and the lower panels denote the SBR in winter. The contours in the Figs. (c, e), (d, f) denote the difference between the SBR in Figs. (c, e), (d, f) with the SBR in Figs. (a, b), respectively."

Yes, Figs. (c, d) and (e, f) show numerical differences to Figs. (a, b).

18. Please add the SBR increments [mW*m-2*sr-1*nm-1] 60.68-20.99=39.69 and 76.36-20.99=55.37 in line 263 because they are listed in the abstract and in the summary.

**Response:** Thanks for your suggestions.

**Original text:** Statistics illustrate that the averaged SBR of the three spaceborne DWLs are 20.99, 60.68, and 76.36 mW·$m^{-2}$·$sr^{-1}$·$nm^{-1}$ respectively.

**Modified:** Statistics illustrate that the averaged SBR of the three spaceborne DWLs are 20.99, 60.68, and 76.36 mW·$m^{-2}$·$sr^{-1}$·$nm^{-1}$ respectively. The increments of averaged SBR received by the new spaceborne DWLs are 60.68-20.99=39.69 mW·$m^{-2}$·$sr^{-1}$·$nm^{-1}$ and 76.36-20.99=55.37 mW·$m^{-2}$·$sr^{-1}$·$nm^{-1}$.

19. The sentence in lines 263-264 ("The quantile statistics of SBR is presented in Table 1, which means that the corresponding percentages of the grids (the earth is divided into 1°×1° grid) of which the SBR will be smaller than the values listed in the first line of Table 1.") is unclear. Please provide a clearer formulation, e.g. also by adding an example (e.g., 90% of the grid points (?) or tiles (?) of the 12:00 orbit have SBR values smaller than 105.77 mW*m-2*sr-1*nm-1).

**Response:** Thanks for your suggestions. An example would make it clear.

**Original text:** The quantile statistics of SBR is presented in Table 1, which means that the corresponding percentages of the grids (the earth is divided into 1°×1°grid) of which the SBR will be smaller than the values listed in the first line of Table 1.

**Modified:** The quantile statistics of SBR is presented in Table 1, which means that the corresponding percentages of the grids (the earth is divided into 1°×1°grid) of which the SBR will be smaller than the values listed in the first line of Table 1. For example, 90% of the grid points of the 12:00 orbit have SBR values smaller than 105.77 mW·$m^{-2}$·$sr^{-1}$·$nm^{-1}$.

20. Please replace "upper layer of troposphere and stratosphere" by "upper layer of atmosphere" in lines 277-278.

**Response:** Thanks for your suggestions.

Because in the lower layer of stratosphere, the wind observation uncertainties would meet the accuracy requirement of ESA. So we think the original expression is more accurate.

21. In the captions of Figs. 3, 5, and 6, the authors write that the "correspondence relationship between the subgraphs and orbits, seasons is consistent with Fig. 2". It is proposed to reformulate this sentence, e.g. to "The arrangement of the subgraphs corresponds to that of Fig. 2".

**Response:** Thanks for your suggestions. We will reformulate the sentence according to your suggestions.

22. In lines 298-299: Is the accuracy level of Aeolus, mentioned here, that one shown in Figs. 3 (a) and (b)? If so, please note this here.

**Response:** Yes, the accuracy level of Aeolus is the one shown in Figs. 3(a) and (b). We will add the expression in the revision.

**Original text:** Supposed that the wind observation accuracy of the two new spaceborne DWLs is required to reach the accuracy level of Aeolus, which can be used for joint observations of the three satellites.

**Modified:** Supposed that the wind observation accuracy of the two new spaceborne DWLs is required to reach the accuracy level of Aeolus as shown in Figs. 3(a, b), which can be used for joint observations of the three satellites.

23. It is assumed that the results shown in Figs. 6 (a) and (b) are identical to those of Figs. 3 (a) and (b). It is however not directly seen due to the different color scales. If so, please make a corresponding note in the text or caption of Fig. 6. If not, please explain why it is not the case.

**Response:** Thanks for your suggestions.

Figs. 6 (a, b) are identical to those of Figs. 3 (a, b). Because both of the laser energies are 60 mJ. We will make a corresponding note in the text in the revision.

**Original text:** The comparison between Fig. 6(c-f) and Fig. 3(c-f) illustrate that, as for the two new spaceborne DWLs, when the laser energy increases from 60 mJ to 80 mJ, the observation accuracy would be improved significantly. (line 357)

**Modified:** Figs. 6(a, b) are identical to those of Figs. 3 (a, b), for that both of them are observed with laser energies of 60 mJ. The comparison between Figs. 6(c-f) and Figs. 3(c-f) illustrate that, as for the two new spaceborne DWLs, when the laser energy increases from 60 mJ to 80 mJ, the observation accuracy would be improved significantly.

24. technical corrections

lines 53-55: Doppler wind lidar which sensing -> senses, Mie/Rayleigh channel sensing -> senses

line 122: expect the mean altitude -> except

Different notations are used for the uncertainty of wind observation in the Rayleigh

channel in Eqs. (1) and (8). Please use a consistent notation.

line 233: the wind observation uncertainty which were calculated -> was

line 454: is also need -> needed

line 463: Subsect. 3.4 does not exist, replace by Subsect. 3.3

**Response:** We're very sorry for the low-level mistakes. And we will correct these mistakes in the revised manuscript.

**References**

AMIRIDIS, V., MARINOU, E., TSEKERI, A., WANDINGER, U., SCHWARZ, A., GIAN-NAKAKI, E., MAMOURI, R., KOKKALIS, P., BINIETOGLOU, I., SOLOMOS, S., HEREKAKIS, T., KAZADZIS, S., GERASOPOULOS, E., PROESTAKIS, E., KOTTAS, M., BALIS, D., PA-PAYANNIS, A., KONTOES, C., KOURTIDIS, K., PAPAGIANNOPOULOS, N., MONA, L., PAPPALARDO, G., Le RILLE, O. ANSMANN, A. (2015), "LIVAS: a 3-D multi-wavelength aerosol/cloud database based on CALIPSO and EARLINET", ATMOSPHERIC CHEMISTRY AND PHYSICS, Vol. 15 No. 13, pp. 7127-7153.

RENNIE, M. (2017) CCN6 results: further Chain-of-Processors testing of L2B results and testing of CCN6 L2B processor algorithm updates., ECMWF.

ZHANG, C., SUN, X., ZHANG, R., ZHAO, S., LU, W., LIU, Y. FAN, Z. (2019), "Impact of solar background radiation on the accuracy of wind observations of spaceborne Doppler wind lidars based on their orbits and optical parameters", Optics Express, Vol. 27 No. 12, pp. A936-A952.

[Figure]

**Fig. 1.** Comparisons between the wind observations of Rayleigh channel and Mie channel. (a)Wind observation uncertainties; (b)useful signal of atmospheric backscatter; (c)signal excited by SBR.

---

## Author Response (AR1)

**Replies to the Reviewers**

Dear reviewers,

We are truly grateful to reviewers' critical comments and thoughtful suggestions. Based on these comments and suggestions, we have made careful modifications on the original manuscript. We are now sending the revised article. Please see our point to point responses to all comments below, and the corresponding revisions in the manuscript. In the revised manuscript, the biggest changes are mainly in the simulation system:

1) New instrument parameters of Aeolus were used according to ADM-Aeolus Algorithm Theoretical Basis Document (ATBD) Level1B products (Reitebuch et al., 2018)

2) The noise of detection unit was considered for that the impact of the noise cannot be negligible.

The comments and suggestions you gave are marked in blue, our modifications are marked in black, the original texts are marked in italic black, and our reasons for the modifications are marked in red. We hope the new manuscript will meet your magazine's standard. Below you will find our point-by-point responses to the reviewers' comments/ questions:

**Reviewer 1:**

**General Comments**

1. The presented results are interesting and potentially useful input for discussions on an Aeolus follow-on mission. The main question, which has not been answered in the paper is: why selecting orbits other than dawn-dusk, given that 3 satellites in a dawn-dusk orbit gives already quite good global coverage, without reducing wind quality due to increased solar background? See Fig.4 of Marseille et al (2008). Is there some indication that different local overpass times would be favorable for NWP? Please elaborate on this in the paper.

**Response:** The main purpose of this manuscript is to access the impact of solar background radiation (SBR) on the accuracy of wind observations for spaceborne Doppler wind lidar (DWL). For spaceborne DWLs operate on sun-synchronous orbits, the dawn-dusk orbit will receive minimum SBR, and the noon orbit will receive maximum SBR. The spaceborne DWLs operate on the sun-synchronous orbits with local time of ascending node (LTAN) crossing of 15:00 will receive medium SBR.

In this paper, the impact of different local overpass times on NWP was not taken into account. The selection of the three orbits was based on the following three aspects:

(1) The solar background radiation received by these three orbits is representative for sun-synchronous orbits.

(2) The observation coverage would be improved with three Aeolus-type spaceborne DWLs operating on the 3 orbits with LTAN crossing of 18:00, 15:00, and 12:00.

(3) We could reconstruct wind diurnal cycle with joint observations of the three spaceborne DWLs.

Based on the three above aspects, paragraph 2 of Introduction was modified:

**Original: Page 2, In. 43**

The future spaceborne DWLs may operate on different orbits which should be related to their observation purposes. Aeolus operates on the sun-synchronous, dawn-dusk orbit to minimize the impact of solar background radiation (SBR) on the accuracy of wind observations (Heliere et al., 2002, Baars et al., 2019). The SBR is defined as the top-ofatmosphere (TOA) radiance which directs to the telescopes of spaceborne DWLs, and the solar background noise (SBN) is the photon counts excited by SBR and imaged on the photon detectors (Zhang et al., 2018) which would lower the observation accuracy by Poisson noise (Liu et al., 2006, Hasinoff et al., 2010). The dawn-dusk orbit is an optimal proposal to lower SBR for spaceborne DWLs operating on sun-synchronous orbits. If the future spaceborne DWLs would operate on the sun-synchronous orbits with different local time of ascending node (LTAN) crossing, the received SBR would become larger which would lead to higher uncertainties of wind observations.

**Modified:**

Aeolus operates on the sun-synchronous, dawn-dusk orbit to minimize the impact of solar background radiation (SBR) on the accuracy of wind observations (Heliere *et al.*, 2002, Baars *et al.*, 2019). The SBR is defined as the top-of-atmosphere (TOA) radiance which directs to the telescopes of spaceborne DWLs, and the solar background noise (SBN) is the photon counts excited by SBR and imaged on the photon detectors (Zhang *et al.*, 2018) which would lower the observation accuracy by Poisson noise (Liu *et al.*, 2006, Hasinoff *et al.*, 2010). The dawn-dusk orbit is an optimal proposal to lower SBR for spaceborne DWLs operating on sun-synchronous orbits. The future spaceborne DWLs may operate on different orbits which should be related to their observation purposes. For example, according to Marseille et al. (2008), larger coverage of wind observations would perform better in improving results of NWP. Furthermore, if the wind field at about 00:00/12:00 or 03:00/15:00 can be observed, we can reconstruct the wind speed diurnal cycle combing with the wind observations of Aeolus. If the future spaceborne DWLs would operate on the sun-synchronous orbits with different local time of ascending node (LTAN) crossing, the received SBR would become larger which would lead to higher uncertainties of wind observations.

2. In connection to this. Line 43: "The future spaceborne DWLs may operate on different orbits which should be related to their observation purposes". Which observation purposes are related to 12:00 or 15:00 local overpass times? See also line 72: "Assuming the future Aeolus-type spaceborne DWLs will operate on the sun-synchronous orbits with different LTAN". Based on what assumption?

**Response:** With joint observations of the three Aeolus-type spaceborne DWLs, we could reconstruct the wind diurnal cycle as mentioned in comment 1.

3. The realism of the simulations can be largely improved by comparing Aeolus measured SBR with simulated Aeolus SBR. Information on Aeolus measured SBR and the impact on Aeolus wind quality is found in the attached supplementary material. Based on this information, the authors can test the realism of their simulations.

**Response:** Thanks very much for providing the material to verify the realism of the simulations.

1) We cannot simulate the measured SBR with our model for that the duration times for background measurements are variable. The highest duration times are 3750  $\mu$ s, while the lowest duration times are 2.1  $\mu$ s. To simulate the measured SBR with one-year range, the corresponding duration times for background measurements are needed which may be contained in the L1A data. However, the L1A data are not openly accessible.

2) The solar background noise (SBN) received by Aeolus is determined by the geometry of solar and satellite, atmospheric conditions, and earth reflectance. And the solar zenith angle (SZA) of the off-nadir points is the main determinants. The simulated SZAs of the off-nadir points within one-year range are shown in Fig. 1. When the SZA is greater than 90 degrees as the horizontal red line shows, the received SBR could be negligible. Comparisons between Fig. 1 (a, b) show high consistence. Both of them are periodic. The values of them reach maximum near summer solstice and reach maximal near winter solstice. And the values of SBR reach minimal values near spring and autumn equinox. The 4 time ranges in Fig. 1 (a) divided by 8 red lines denote 15 days near autumn equinox, winter solstice, spring equinox, summer solstice.

Fig. 1. (a) The simulated solar zenith angle of the off-nadir points of Aeolus within one-year range.

(b) Orbital variation of Rayleigh solar background noise (obtained from supplement).

3) The received SBR within 15 days near summer and winter solstice was simulated. And the SBR in summer and winter solstice was converted to ACCD counts of Rayleigh channel as is illustrated in Fig. 2. As Fig. 2 illustrated, the amount of ACCD counts near summer and winter solstice are consistent with Fig. 1 of the supplement. And SBN excited on Rayleigh channel are periodic as the subgraph of the supplement shows.

Fig. 2. The received SBN of Rayleigh channel (A+B) within 1 day. (a) simulated SBN near summer solstice, (b) simulated SBN near winter solstice, (c) measured SBN.

4) Assuming under the same atmospheric conditions, how much difference will the wind observations uncertainties obtained by using the new and old parameters be? The simulated uncertainties of wind observations under cloud-free atmospheric condition are illustrated in Fig. 3. The corresponding SZA of Fig. 3 is 70°, and the related SBR is 72.19 mW·m-2·sr-1·nm-1. As Fig. 3 (a) shows, the difference of uncertainties simulated using old and new parameters is large. The largest and average difference is 2.17 and 0.61 m/s, respectively. Fig. 3 (b) illustrated that the SNR simulated using new parameters is obviously large than the results using old parameters. The combination of Figs. 3 (c, d) can account this phenomenon. In Fig. 3 (c), the simulated useful signal of Rayleigh channel is relatively close, almost no difference. However, Fig. 3 (d) illustrated that the simulated SBN obtained using old parameters is much large than that of new parameters. Why the SBN simulated using old parameters is much larger than that of new parameters is much larger than that of the new parameters. And in the simulation process, the SBN is regarded as following uniform distribution. Wider transmission bandwidth of Rayleigh channel will lead to higher solar background energy, which would lower the SNR of Rayleigh channel, and then increase the uncertainties of wind observations.

Considering the difference in simulated wind observation uncertainties between the old and new parameters. In the modified manuscript, new instrument parameters were used.

---

## Referee Report (RR1)

re-review of the revised version amt-2020-202-manuscript-version3.pdf

In my first review, I had addressed a couple of issues. The authors responded to all of them adequately in their present revised version. In particular, they have recomputed their results by using the recent satellite and instrument parameters reported in the ATBD issue 4.4, 20.04.2018, by O. Reitebuch et al. In this way, the usefulness of the investigations and results reported by the authors has been substantially increased. Thus it is proposed to publish the paper in AMT. Before doing so, some spelling errors should be corrected.

Additionally, I would like to comment on the author's reply to my former item (7):

Of course, the Mie channel is used to detect winds in the PBL. In their original paper however, the authors showed also the required laser pulse energy down to the PBL, based on their investigations in the Rayleigh channel. So it was interesting for me to get the author's opinion on the observed behaviour of the required laser pulse energy in the PBL. In their reply now, the authors do not want to speculate about this behaviour and mention that the accuracy of the Rayleigh channel winds in the PBL is not considered in the paper. I accept their point of view. To my opinion, backscattered signals of aerosols and clouds, present in the PBL, are also measured in the Rayleigh channel (cross talk from the Mie channel) due to an imperfect filtering. This leads to a larger signal level in the Rayleigh channel than it would be case without any clouds and aerosols. Thus the signal to noise ratio is also better, the wind uncertainties decrease, and the required laser pulse energy to meet a specified accuracy criterion decreases.

Furthermore, the authors raised the question why the wind observation accuracy in the Mie channel is higher than that of the Rayleigh channel, in the PBL. In the PBL, the aerosol and cloud particles produce strong backscattered signals which can be seen as sharp peaks in the spectrum. The corresponding Doppler shifts can be determined more accurately than those of the broader molecular spectra. Consequently, the Mie channel wind uncertainties are smaller than those of the Rayleigh channel.

---

## Author Response (AR2)

Dear Dr. Ad Stoffelen,

Thanks very much for yours and other reviewers' work in improving the manuscript. We are truly grateful to yours and other reviewers' critical comments and thoughtful suggestions. Based on these comments and suggestions, we have made careful modifications on the original manuscript. We are now sending the revised manuscript. The comments and suggestions you and reviewers gave are marked in blue, our modifications are marked in black, the original texts are marked in italic black, and our reasons for the modifications are marked in red. Below you will find our point-by-point responses to your and the reviewers' comments/ questions:

**AE's comment**

Overall, the reviewers are pleased with your responses to their comments, representing clarification of the manuscript and its value for AMT. Nevertheless, the requirement for needs to be clearly expressed, as well as the degradation by not measuring in dawn-dusk. Please clarify the inconsistencies found by the reviewer. Finally and importantly, for work by your colleagues, the use of the English language needs improvement and corrections by a fluent colleague in English are highly recommended.

**Response:** In the modified manuscript, we described the requirements for the Aeolus operating on the other sun-synchronous orbits according to Referee 3's suggestions. The degradation in wind observation uncertainties by measuring in other orbits and the inconsistencies between our rebuttal and the referee's supplementary material were clarified in this document. The detail can be found in the response to Referee 3's comment. Moreover, our manuscript was modified by native English speaker. We hope the new manuscript will meet your magazine's standard.

**Referee 2's comment**

In my first review, I had addressed a couple of issues. The authors responded to all of them adequately in their present revised version. In particular, they have recomputed their results by using the recent satellite and instrument parameters reported in the ATBD issue 4.4, 20.04.2018, by O. Reitebuch et al. In this way, the usefulness of the investigations and results reported by the authors has been substantially

increased. Thus it is proposed to publish the paper in AMT. Before doing so, some spelling errors should be corrected.

**Response:** Thanks again for your previous comments. They did much help in improving the manuscript. In this modification, the manuscript was modified by native English speaker, we think most spelling errors should have been corrected.

Additionally, I would like to comment on the author's reply to my former item (7):
Of course, the Mie channel is used to detect winds in the PBL. In their original paper however, the authors showed also the required laser pulse energy down to the PBL, based on their investigations in the Rayleigh channel. So it was interesting for me to get the author's opinion on the observed behaviour of the required laser pulse energy in the PBL. In their reply now, the authors do not want to speculate about this behaviour and mention that the accuracy of the Rayleigh channel winds in the PBL is not considered in the paper. I accept their point of view. To my opinion, backscattered signals of aerosols and clouds, present in the PBL, are also measured in the Rayleigh channel (cross talk from the Mie channel) due to an imperfect filtering. This leads to a larger signal level in the Rayleigh channel than it would be case without any clouds and aerosols. Thus the signal to noise ratio is also better, the wind uncertainties decrease, and the required laser pulse energy to meet a specified accuracy criterion decreases.

**Response:** Thank you for your further discussion with me on this questions. We understand your points, and will carry out related simulation research in the next step, and further communicate with you if any results are obtained.

Furthermore, the authors raised the question why the wind observation accuracy in the Mie channel is higher than that of the Rayleigh channel, in the PBL. In the PBL, the aerosol and cloud particles produce strong backscattered signals which can be seen as sharp peaks in the spectrum. The corresponding Doppler shifts can be determined more accurately than those of the broader molecular spectra. Consequently, the Mie channel wind uncertainties are smaller than those of the Rayleigh channel.

**Response:** Thanks very much for your answer about the question.

**Referee 3's comment**

The authors provide suggestions for future Aeolus-type follow-on missions, with different local overpass times compared to 6/18 UTC (dawn-dusk) for Aeolus and taking into account increased solar background radiation in measured Aeolus signals, hence reduced data quality, as a consequence of selecting different sun-synchronous orbits.

I thank the authors for considering my earlier review and corresponding comments to improve the manuscript.

Some aspects are still not convincing enough in my view.

point 1

====

line 10: "For that the future spaceborne DWLs may not operate on sun-synchronous dawn-dusk orbits due to their observation purposes"

The "observation purposes" have not been mentioned in the text. The authors seem to suggest in lines 52-53 that being more flexible on orbit selection for an Aeolus follow-on (FO) mission, rather than fixed to dawn-dusk as in Marseille (2008), offers the possibility to sample the diurnal cycle with Aeolus.

The motivation or need for Aeolus-FO to sample the diurnal cycle is not given in the text. The authors could refer here to the WMO OSCAR database which provides a list of requirements for future observing systems to be beneficial for NWP, among others (http://www.wmo-sat.info/oscar/requirements, see Ids 311-313). Aeolus meets the observation cycle threshold requirement of 12 hours. The orbits suggested by the authors improve on this, approaching the "breakthrough" requirement.

Another motivation for flying other than dawn-dusk is experience from scatterometer use in global NWP. Scatterometers measure winds near the ocean surface. It has been demonstrated that scatterometers provide independent information to NWP for overpass times separated by only ~3 hours (Indian scatterometer OSCAT with ~12UTC local overpass time provides independent information relative to

ASCAT with ~9:30 local overpass time). Some details are found in figure 24 of the pdf at https://www.knmi.nl/kennis-en-datacentrum/publicatie/research-and-development-in-europe-on-global-application-of-the-oceansat-2-scatterometer-winds

**Response:** Thanks for your suggestions. In the modified manuscript, we added the motivations of Aeolus-FO flying on the other orbits in Section Introduction.

**Original:** Ln 37, Page 2

*In addition, Marseille et al. (2008) demonstrated that larger observation coverage is more beneficial in the improvement of NWP results in global scale compared to the measurement of horizontal vector wind by proposing several multi-satellites joint observation scenarios with Aeolus-type instruments. However, the measurements of horizontal vector wind perform better for NWP results in the region close to the satellite tracks.*

**Modified:** Ln 42, Page 2

In addition, Marseille et al. (2008) demonstrated that a larger observation coverage is more beneficial in the improvement of NWP results on global scale compared to the measurements of the horizontal vector wind by proposing several multi-satellite joint observation scenarios with Aeolus-type instruments. Regarding multi-satellite joint observation scenarios, according to the World Meteorological Organization's (WMO) Observing Systems Capability Analysis and Review Tool (OSCAR) (Eyre, 2009), an observation cycle of 12 h with Aeolus operating on a sun-synchronous dawn-dusk orbit would meet "the minimum" requirements that have to be met to ensure the observations are useful for global NWP. When another Aeolus-type satellite operates on a sun-synchronous noon-midnight orbit combined with Aeolus, the observation cycle may become 6 h, which would meet breakthrough requirement that, if achieved, would result in a significant improvement in global NWP compared with those based on a single Aeolus.

**Original:** Ln 43, Page 2

*The future spaceborne DWLs may operate on different orbits which should be related to their observation purposes. For example, according to Marseille et al. (2008), larger coverage of wind observations would perform better in improving results of NWP. Furthermore, if the wind field at about 00:00/12:00 or*

*03:00/15:00 can be observed, we can reconstruct the wind speed diurnal cycle combing with the wind observations of Aeolus. If the future spaceborne DWLs would operate on the sun-synchronous orbits with different local time of ascending node (LTAN) crossing, the received SBR would become larger which would lead to higher uncertainties of wind observations.*

**Modified:** Ln 55, Page 2

Future spaceborne DWLs may operate on different orbits according to their observation purposes. According to experience gained from scatterometers used in global NWP (Stoffelen et al., 2013), it has been demonstrated that the forecasting errors of tropical cyclone positions are much lower when the Indian Space Research Organisation's (ISRO) scatterometer, which has an ~12:00 UTC local overpass time, is assimilated in the NWP with the original METOP-A and METOP-B (~9:30 UTC local overpass time). Therefore, it is assumed that if the global wind field at about 00:00/12:00 or 03:00/15:00 can also be observed, the global forecast may also be significantly improved.

point 2

====

I am not convinced that the simulations of SBR are representative for Aeolus. Figure 2c of the rebuttal shows values up to 5e+5 ACCD counts, while Figure 2b (also winter period) shows values a factor of 10 higher. How can the authors conclude that "the number of ACCD counts is consistent"?

**Response:** Although the solar background noise (SBN) shown in Figure 2b and Figure 2c of the rebuttal are both in winter period, Figure 2b shows the peak values of SBN in winter and Figure 2c shows the valley values in winter as Figure 1a and 1b of the rebuttal illustrate. And Figure 1b indicates that the peak value of SBN in winter is likely to be 10 times the valley values.

In the rebuttal, the sentence "As Fig. 2 illustrated, the amount of ACCD counts near summer and winter solstice are consistent with Fig. 1 of the supplement." is derived from the fact that:

1) The amount of global maximum and local maximum values of SBN near summer and winter solstices are about 1e7 and 0.5e5, and that are consistent with the values shown Figure 1b of rebuttal.

2) From Figure 2(a, b) of the rebuttal, the values of SBN show periodic variation, which are consistent with the periodic variation of Figure 2c.

Regarding Figure 4 of the rebuttal and figure 2 of the supplement. First the authors conclude: "The comparisons between Fig. 4 and Fig. 2 of the supplement show large difference." Can the authors explain this large difference?

[Figure]

Fig. 1. The relationship between uncertainties of wind observations and useful signal of channel A on Rayleigh channel. (a) Figure 4 of the rebuttal; (b) Fig. 2 of the supplement.

**Response:** As is illustrated in Fig. 1(a), the same as Fig. 4 of the rebuttal, when useful signal on channel A reach 5000, the uncertainty on Rayleigh channel is about 8 m/s on typical solar background noise. However, the uncertainty is about 4 m/s when useful signal on channel A is 5000 on Fig. 1(b), the same as Fig. 2 of the supplement. The fact demonstrates that "the comparisons between Fig. 4 and Fig. 2 of the supplement show large difference".

"However, the uncertainties of wind observation are about 2~3 m/s when the SBR is about 72.19 mW·m−2·sr−1·nm−1". The value of 72.19 corresponds to a "typical" SBR value, which is plotted as a red curve in Figure 4. I cannnot see uncertainly values of 2-3 m/s in this plot. Can the authors please explain?

[Figure]

Fig. 2. HLOS wind uncertainties of Rayleigh channel for three level of solar background noise under clear skies. (a) Results of TN17.4 (Figure 4) (Rennie, 2017), the blue lines show the median absolute deviation of the error, and the three lines show uncertainties under no solar background radiation (SBR), typical SBR, and worst SBR, respectively; (b)The simulation results of our model.

**Response:** (Rennie, 2017) have tested the wind uncertainties of Rayleigh channel for three level of solar background radiation (SBR): no SBR, typical SBR (72.19 mW·m$^{-2}$·sr$^{-1}$·nm$^{-1}$), and worst SBR (154 mW·m$^{-2}$·sr$^{-1}$·nm$^{-1}$, polar summer) as the blue lines of Fig. 2(a) shown. Fig. 2(a) illustrates that the wind uncertainties of Rayleigh channel is relatively small in the free troposphere, and large in planet Boundary Layer (PBL) and stratosphere; the differences of wind observation uncertainties in the free troposphere are not large under the three level of SBR, all of them are about 2~3 m/s. Fig. 2(b) which was simulated by our model shows the similar characteristics. The sentence "he uncertainties of wind observation are about 2~3 m/s when the SBR is about 72.19 mW·m$^{-2}$·sr$^{-1}$·nm$^{-1}$" refers to the uncertainties in the free troposphere. The comparisons between the results of Rennie (2017) and our simulations verify the rationality of our model.

In the rebuttal the authors state: "In the manuscript, Aeolus was assumed to be operated on best case scenario." Please add this to the abstract explicitly, to ensure that the context of the manuscript is very clear for all readers already at the beginning, also for those readers who are used to work with real Aeolus data, whose random error is substantially worse than the best case scenario presented in the manuscript.

**Response:** In the modified manuscript, the related descriptions are added in the abstract.

**Original:** Ln 13, Page 1

*On the two new orbits, the increments of averaged SBR received by the new spaceborne DWLs range from 39 to 56 mW·m$^{-2}$·sr$^{-1}$·nm$^{-1}$ under cloud-free skies near summer and winter solstices, which will lead to the increment of averaged Rayleigh channel wind observation uncertainties of 0.19 m/s for 15:00 orbit and 0.27 m/s for 12:00 orbit using the instrument parameters of Aeolus with 30 measurements per observation with 20 laser pulses per measurement.*

**Modified:** Ln 13, Page 1

On these two new orbits, the increments of the averaged SBR received by the new spaceborne DWLs range from 39 to 56 mW·m$^{-2}$·sr$^{-1}$·nm$^{-1}$ under cloud-free skies near the summer and winter solstices, which will lead to uncertainties of 0.19 m/s and 0.27 m/s in the increment of the averaged Rayleigh channel wind observations for 15:00 and 12:00 orbits using the instrument parameters of Aeolus with 30 measurements per observation and 20 laser pulses per measurement. This demonstrates that Aeolus operating on the sun-synchronous dawn-dusk orbit is the optimal observation scenario, and the random error caused by the SBR will is larger on other sun-synchronous orbits.

Line 316: "The comparison illustrates that SBR caused the maximum increase in the averaged wind observation uncertainty of about 3.04-2.61=0.43 m/s for Aeolustype DWLs operating on the sun-synchronous orbits."

That is actually a marginal degradation, so that could be an argument to fly other than dawn dusk, in case of flying more than a single Aeolus type instrument at the same time. These random error values correspond to the right hand side of the last figure in the supplementary material, which shows an increase of 1.2 m/s random error between typical and worst case SBR scenario, so a factor of 3 larger increase than simulated by the authors. It seems that the simulated results of the authors are much more positive than what can be infered from real Aeolus data. Can the authors please comment?

**Response:** The wind observation uncertainties caused by SBR are influenced by not only the amount of SBR, but also the photon number backscattered by atmospheric molecules. When the photon number excited by the atmospheric molecules is large enough, the wind observation uncertainties will be small

even if the SBR is large enough. And when the photon number excited by the atmospheric molecules is small, little SBR would cause large wind observation uncertainties.

When the useful signal on channel A is up to 10000, the difference of wind observation uncertainties between typical and worst SBR is 1.37 m/s in our simulation, as is shown in Fig. 1(a). The result is close to the result in the supplementary material.

The average SBR received by the three orbits are 20.09, 60.68, and 76.36 $mW \cdot m^{-2} \cdot sr^{-1} \cdot nm^{-1}$ respectively. The SBR are not large compared to worst case (154 $mW \cdot m^{-2} \cdot sr^{-1} \cdot nm^{-1}$). Furthermore, the useful signals of Rayleigh channel are 2.65e+4 each bin in the free troposphere and stratosphere for the three orbits. The useful signals are 2.5 times the useful signals of the supplementary material. The two facts can account for the little differences of averaged uncertainties among the three orbits. According to Fig. 2(a) derived from the TN 17.4, the differences among the three SBR conditions are not large, especially in the free troposphere.

**References:**

RENNIE, M. (2017) TN17.4 CCN6 results: further Chain-of-Processors testing of L2B results and testing of CCN6 L2B processor algorithm updates., ECMWF.

STOFFELEN, A., VERHOEF, A., VERSPEEK, J., VOGELZANG, J., MARSEILLE, G., DRIESENAAR, T., RISHENG, Y., De CHIARA, G., PAYAN, C., COTTON, J., BENTAMY, A. & PORTABELLA, M. (2013) Research and Development in Europe on Global Application of the OceanSat-2 Research and Development in Europe on Global Application of the OceanSat-2 Scatterometer Winds: Final Report of OceanSat-2 Cal/Val AO project., KNMI, Royal Netherlands Meteorological Institute, de Bilt, the Netherlands.

---

## Author Response (AR3)

Dear Dr. Ad Stoffelen,

Thank you for taking time out of your busy schedule to review our manuscript. In this revision, we made careful modifications based on your comments and suggestions. After these rounds of revision, your critical comments help us a lot in improving the manuscript, we're really appreciate for your help. We are now sending the revised manuscript. Please see our point to point responses to all your comments below.

In this letter, the comments and suggestions you gave are marked in blue, our modifications are marked in black, the original texts are marked in italic black, and our explanation for the modifications are marked in red.

**AE's comments:**

Thanks for this complete rebuttal and modifications, which indeed render the manuscript acceptable for publication in AMT.

I suggest a few minor more corrections to improve the clarity of the manuscript.

**Response:** We re-read the paper carefully and made corresponding modifications to some ambiguous places. For example:

**Original:** Ln 264, P10:

*The contours in Fig. 3 denote the differences between the SBRs of the two new orbits and sun-synchronous dawn-dusk orbit, which demonstrates that the dawn-dusk orbit is the optimal observation scenario for minimizing received SBR for Aeolus-type spaceborne DWLs operating on sun-synchronous orbits.*

**Modified:** Ln 263, P10:

The contours in Fig. 3 denote the differences between the SBR of the two new orbits and sun-synchronous dawn-dusk orbit. All the values are positive, indicating that the dawn-dusk orbit is the optimal observation scenario for minimizing received SBR for Aeolus-type spaceborne DWLs operating on sun-synchronous orbits.

**Original:** Ln 457, P18:

*The required energy is determined by temperature, pressure, wind uncertainty, SBR, and noise of instrument, and thus, the required laser pulse energies are different in different bins.*

**Modified:** Ln 458, P18:

According to this method, the required energy is based on the temperature, pressure, wind uncertainty, SBR, and noise of the instrument, and thus, the required laser pulse energies are different in different bins.

Some unclear sentences that have nothing to do with the subject of the manuscript were deleted:

**Original:** Ln 301, P11:

*4) Compared with other regions, the uncertainties in the equatorial region are higher at the bottom of the troposphere and are lower in the stratosphere. The trend of the temperature profile in the equatorial region is the main reason for this phenomenon, which is consist with the trend of the uncertainties. The number density of molecules is inversely proportional to the temperature. A low molecular number density leads to a weak return signal of spaceborne DWLs, which leads to higher wind observation uncertainties.*

**Modified:** Ln 302, P11:

4) Compared with other latitudinal regions, the uncertainties in the equatorial region are higher at the bottom of the troposphere and are lower in the stratosphere.

**Original:** Ln 313, P11:

*As can be seen by comparing between Fig. 4 and 5, the wind observation uncertainties become larger as the impact of the SBR increases. The uncertainties exhibt obvious latitudinal variations. This is mainly attributed to the latitudinal variations in the maximum SBR shown in Fig. 3.*

**Modified:** Ln 311, P11:

As can be seen by comparing between Fig. 4 and 5, the wind observation uncertainties become larger as the impact of the SBR increases.

Some expressions have been added to make the meaning of the manuscript more clearly.

**Original:** Ln 324, P12:

*This small degradation of the uncertainties could also be used as an argument for operating Aeolus-type spaceborne DWLs on other sun-synchronous orbits rather than a dawn-dusk orbit.*

**Modified:** Ln 322, P12:

This small degradation of the uncertainties could also be used as an argument for operating Aeolus-type spaceborne DWLs on other sun-synchronous orbits rather than a dawn-dusk orbit, in case of flying more than a single a single Aeolus-type instrument at the same time.

And some other minor corrections.

The new paragraph quoting Marseille et al (2008) ends with "a single Aeolus". However, Marseille et al. showed that up to three Aeolus satellites in an dawn-dusk orbit are effectively contributing to NWP benefits. Please replace the quoted text with "dawn-dusk Aeiolus".

**Response:** Done.

Some of the red text in reply to the second reviewer may be promoted to the manuscript text for consideration by its readers.

**Response:** In the modified manuscript, two points in the reply to the reviewers were added.

**Point 1** (Ln 327, P12):

According to Rennie (2017), the worst case SBR (154 mW m$^{-2}$ sr$^{-1}$ nm$^{-1}$, polar summer condition) has noise around 0.5~1.0 m/s lager random error than the best case (night-time condition) at the height of 5~10 km. This result illustrates the degradation of the uncertainties in Rayleigh channel is not large in troposphere and also indicates the correctness of the increase in wind observation uncertainty between different orbits calculated in this study.

The above expression was added. In the above, we get the result that the new Aeolus-type spaceborne DWLs operating on the two new orbits would get a small degradation in average wind observation uncertainties compared to that of Aeolus. To show the correctness of the result, the results in Rennie (2017) was added, which was also expressed in the reply to the second reviewer in former rebuttal.

**Point 2 :**

Thanks to the first reviewer for explaining the question why the wind observation of Mie channel is more accurate than that of Rayleigh channel in PBL. We cited the reason in the modified manuscript.

**Original** (Ln 291, P11):

*In fact, the Mie channel is mostly used for wind observations due to the widespread presence of aerosols in the PBL. Therefore, the accuracy of the Rayleigh channel in the PBL is not considered in the following section of this paper.*

**Modified** (Ln 289, P11):

In fact, the Mie channel is mostly used for wind observations due to the widespread presence of aerosols in the PBL. Because the aerosols produce strong backscattered signals which can be seen as sharp peaks in the spectrum. The corresponding Doppler shifts can be determined more accurately for the spectra of sharp peaks than those of the broader molecular spectra received by Rayleigh channel. Consequently, the Mie channel wind uncertainties are smaller than those of the Rayleigh channel. Therefore, the accuracy of the Rayleigh channel in the PBL is not considered in the following section of this paper.

Line 20: will is -> will be

**Response:** Done.